# FEDERATED DATA AND FEATURE SELECTION BY GENERALIZED CUR DECOMPOSITION

## ABSTRACT

With the advance of federated learning (FL) in privacy-sensitive domains such as healthcare, finance, and mobile intelligence, the need for efficient and robust training becomes increasingly urgent. Communication bottlenecks, heterogeneous client distributions, and fairness requirements make it essential to select the "right" data and features for model training. Yet existing FL research often addresses feature selection and data selection separately, ignoring their interplay in real-world high-dimensional and noisy datasets, leading to suboptimal performance. In this paper, we propose a unified framework for data and feature selection by formulating the problem as a generalized CUR decomposition problem. We introduce FedGCUR, a practical framework that integrates a federated column-pivoted QR (FedCPQR) decomposition routine with per-silo row selection. Specifically, FedCPQR is designed to securely compute a global pivot order without exposing raw data, while FedGCUR leverages this to jointly select shared features and silo-specific samples. We prove that FedCPQR produces exactly the same decomposition results as centralized CPQR and establish an upper bound of the reconstruction error of FedGCUR. Extensive empirical results show that the proposed framework achieves higher accuracy compared to the baselines of data and feature selection methods, demonstrating its effectiveness and efficiency.

## 1 INTRODUCTION

Federated learning (FL) has emerged as a key paradigm for collaborative model development across data silos and edge devices, enabling training without centralizing sensitive records (Zhang et al., 2021; Kairouz et al., 2021). By joint local learning and global model aggregation, FL leverages the knowledge of distributed clients in a privacy-preserving manner (Li et al., 2020). However, realizing these benefits in practice is non-trivial: statistical heterogeneity across silos (Ye et al., 2023), limited communication budgets (Niknam et al., 2020), and the risk of information leakage (Truex et al., 2019) during intermediate computations all complicate the training process. Addressing these issues often requires improving the efficiency and robustness of federated training, not only by designing better optimization protocols, but also by carefully selecting which data and features participate in the learning process.

Recently, numerous studies in federated learning have investigated data and feature selection (Fu et al., 2023; Hu et al., 2023; Wang et al., 2023). The main idea is to select a compact set of globally shared features to reduce client-side computation and uplink communication costs. Moreover, under participation or communication constraints, selecting more informative samples can further improve model prediction and distillation performance. Existing studies typically consider feature selection (Fu et al., 2023; Zhang et al., 2023; Hermo et al., 2024) and data selection (Liu et al., 2022; Li et al., 2021; Rizk et al., 2022) as isolated subproblems rather than as interdependent decisions within a unified framework. Moreover, they often simplify the problem by directly transferring centralized techniques to cross-silo scenarios, without fully accounting for the heterogeneity inherent in FL. However, feature and data selection are coupled in many applications (Garcia-Pedrajas et al., 2021; Dornaika, 2021; Fan et al., 2022), e.g., in a high-dimensional datasets with many noisy and redundant data points. In such settings, if one axis is optimized in isolation, the result can be suboptimal or even contradictory.

One of the highly related techniques for data and feature selection is the matrix decomposition (Lu, 2022), which is fundamental to machine learning and data analysis (Caiafa et al., 2020; Sidiropoulos et al., 2017; Li et al., 2018). They are useful in dimension reduction, principal analysis, coreset discovery, etc. Unfortunately, most methods assume centralized access to the full matrix and thus do not directly apply in FL due to privacy and bandwidth constraints. Recent work has made preliminary progress on federated matrix decomposition (Chai et al., 2022; Hartebrodt & Röttger, 2023) by combining secure aggregation and data masking to avoid exposing raw data, but more expressive federated matrix decompositions remain underexplored.

Among decomposition-based selection methods, CUR is particularly appealing because it directly selects columns and rows of the data matrix Mahoney & Drineas (2009); Li et al. (2018). Given a matrix $A$, CUR seeks factors $A \approx CMT$, where $C$ collects a subset of columns (features), $T$ collects a subset of rows (examples), and $M$ is a small linking matrix. In FL, however, CUR must satisfy federated constraints that do not arise centrally: (i) select a shared set of features so that a joint model can be trained and aggregated across silos; (ii) reveal only privacy-preserving aggregated statistics. These requirements make the federated CUR problem challenging, because rank-revealing selection typically needs global coordination; and heterogeneous silos may prefer different pivots; and naive protocols either leak information or incur prohibitive costs.

In this paper, we bridge this gap by formulating federated data and feature selection as a federated generalized CUR (FedGCUR) decomposition problem. To the best of our knowledge, this is the first work that considers both data and feature selection in FL setting. By designing FedGCUR, a practical algorithmic framework that satisfies the above constraints, our approach has two components. First, we develop a federated column-pivoted QR decomposition routine (FedCPQR) that computes a global column pivot order using only secure sums of squared norms and inner products, with no raw data sharing. Second, we build FedGCUR on top of this routine: it uses the global pivots to select a common set of columns and then performs per-silo row selection locally, yielding CUR factors that approximate each silo's data. The overall design is communication and computation efficient, compatible with standard secure aggregation, and well-suited to FL setting. Theoretical analyses are conducted to show that the proposed FedCPQR outputs exactly the same results as the centralized modified Gram–Schmidt based CPQR, and maintains the same level of privacy as FedQR (Hartebrodt & Röttger, 2023). Extensive experiments are conducted to validate the correctness, effectiveness and efficiency. The results show that FedGCUR achieves higher accuracy compared to the baselines of data and feature selection methods.

## 2 RELATED WORKS

**Federated data selection and feature selection**  Recent work explores privacy-preserving data and feature selection in federated settings (Fu et al., 2023; Zhang et al., 2023; Li et al., 2021; Rizk et al., 2022). For example, Hu et al. (2023) propose a heuristic federated feature selection scheme where each client performs local subset selection and shares only the selected feature indices and their local accuracy, after re-evaluating others' subsets, and the aggregator selects a global subset via an aggregation rule. Wang et al. (2023) propose FAST, a federated learning framework that jointly adapts class-wise data sampling and the number of local iterations to counter Non-IID data and device heterogeneity, with an online MAB-based controller and a proved convergence bound. Hermo et al. (2024) propose Fed-mRMR, a lossless federated adaptation of the classic mRMR filter that replaces raw-data sharing with an occurrences matrix built from local bitmaps and merged across clients, preserving the exact mRMR ranking under non-IID, cross-silo FL. Most existing methods focus on either data selection or feature selection in isolation, which can be suboptimal when both are required, as they ignore the coupling between instances and features.

**Federated matrix decomposition**  Matrix decomposition in federated learning receives less attention. Strong privacy constraints preclude sharing matrix statistics, making standard decompositions difficult to adapt. Hartebrodt & Röttger (2023) studied federated QR decomposition. A modified Gram-Schmidt based QR decomposition method is proposed for federated learning settings. This method approximates $[A_1, \ldots, A_s]^\top \approx [Q_1, \ldots, Q_s]^\top R$, where each client $c$ can only access its corresponding local matrix $Q_c$. The right triangular matrix $R$ is shared globally. Specifically, each site locally maintains its rows and iteratively performs Gram-Schmidt steps, with the only cross-site communication being the sums of inner products and norms. An additive aggregation protocol is

employed to securely aggregate the necessary scalars. Chai et al. (2022) investigate FedSVD for vertical FL. A trusted authority generates a unitary random mask, clients upload masked data, and the server computes an SVD on the masked matrix under secure aggregation. Each client receives the right singular vectors corresponding to its features, while the left singular vectors and singular values are shared globally. Tensor decomposition in FL (Gao et al., 2021; Ouyang et al., 2024) has also been explored, but it lies beyond the scope of this paper.

## 3 METHODOLOGY

### 3.1 PRELIMINARIES

**Notations** In this paper, matrices or sets are denoted by capital letters, vectors by bold lowercase letters, and scalars by lowercase letters. We use superscript $\dagger$ to represent the pseudo-inverse of a matrix, and $\sigma_j$ represents the $j$-th largest singular value. An $m \times m$ identity matrix is denoted by $I_m$, and we omit the subscript when the context is clear. Denote by $\| \cdot \|_2$ the spectral norm for matrix and $\ell_2$ norm for vector, and $\| \cdot \|_F$ the Frobenius norm. We define $[n] = \{1, 2, \ldots, n\}$.

This work focuses primarily on cross-silo horizontal FL scenarios. Specifically, given a data matrix $A \in \mathbb{R}^{n \times d}$, where each row corresponds to data from $s$ different sites, we have $A = [A_1, \ldots, A_s]^\top$, with $A_c \in \mathbb{R}^{n_c \times d}$, and $\sum_{c=1}^s n_c = n$. Here, $c$ represents the client index. We assume that each row of $A$ is randomly sampled from an unknown distribution, though the split function across clients is arbitrary. The target vector is denoted as $\boldsymbol{b} \in \mathbb{R}^{n \times 1}$. Throughout, $k$ is the target column rank, $j$ represents the column index, and $r_c$ is the number of selected data point in silo $c$, with $r = \sum_{c=1}^s r_c$ and $r_c \leq n_c$.

**Secure Aggregation Protocol** We follow Hartebrodt & Röttger (2023) to use the additive secure aggregation protocol introduced by Cramer et al. (2015), which, for a set of scalar values $\{x_c\}_{c=1}^s$, securely returns the sum $\sum_{c=1}^s x_c$ to all participants without revealing the individual $x_c$ values. This is mainly achieved by introducing a large prime known to all participants and transmitting the remainder of the data to be aggregated.

**CUR and Generalized CUR Decomposition** CUR decomposition seeks a subset of $k$ rows and columns from $A \in \mathbb{R}^{n \times d}$ such that $A \approx CMT = ADMS^\top A$, where $C \in \mathbb{R}^{n \times k}$ and $T \in \mathbb{R}^{k \times d}$ are sub-matrices of the columns and rows of $A$, respectively. Here, $D \in \mathbb{R}^{d \times k}$ and $S \in \mathbb{R}^{n \times k}$ are *index selection matrices*, constructed from columns of the identity matrix $I$ with appropriate permutations.

Generalized CUR decomposition is usually defined on a pair of matrices (Gidisu & Hochstenbach, 2022; Cao et al., 2024). Here, we generalize its definition to $s$ matrices: $A_1, \ldots, A_s$. The formal definition is as follows:

**Definition 1** (Generalized CUR Decomposition for Multiple Matrices). *Let $s \in \mathbb{N}, s > 1$, and $c \in [s]$, $\{A_c\}_{c=1}^s$ be a collection of matrices where $A_c \in \mathbb{R}^{n_c \times d}$, $n_c \geq d$. Given the number of sampled rows $r_c$ for $A_c$ (with $r_c \leq n_c$) and the number of sampled columns $k$ (with $k \leq d$), the generalized CUR decomposition of $\{A_c\}_{c=1}^s$ seeks approximations of the form*

$$A_c \approx C_c M_c T_c = A_c D M_c S_c^\top A_c, \tag{1}$$

*where, $S_c \in \mathbb{R}^{n_c \times r_c}$ and $D \in \mathbb{R}^{d \times k}$ are index selection matrices. When $s = 2$, the decomposition problem coincides with Gidisu & Hochstenbach (2022).*

It is important to note that all $\{A_c\}_{c=1}^s$ share the same index selection matrix $D$ in the generalized CUR decomposition.

### 3.2 FEDERATED CPQR

Considering federated data and feature selection, it is necessary to perform feature selection globally without exposing the raw data held by each client. This task can be formulated as a federated column-pivoted QR decomposition problem. The primary difference between standard QR (Watkins, 1982) and column-pivoted QR (CPQR) (Quintana-Ortí et al., 1998) lies in the column selection strategy. Specifically, let $U = [U_1, \ldots, U_s]^\top$ be the working matrix. At iteration $i$,

---

**Algorithm 1** FedCPQR

---

**Input:** Each site $c \in [s]$ holds $A_c \in \mathbb{R}^{n_c \times d}$.
1: **Initialize:** $J \leftarrow [d]$, $P \leftarrow I_d$, $R \leftarrow 0_{d \times d}$. $U_c \leftarrow A_c$, for $c \in [s]$.
2: For each $j \in J$, $c \in [s]$: compute $d_{j,c} \leftarrow \|U_c(:,j)\|_2^2$; $d_j \leftarrow \text{SECAGG}(\{d_{j,c}\}_{c=1}^s)$.
3: **for** $i = 1$ **to** $d$ **do**
4:     $p \leftarrow \arg\max_{j \in J} d_j$. Swap columns $i \leftrightarrow p$ in $P$, in each local $U_c$, and in the first $i-1$ rows of $R$; update $J \leftarrow J \setminus \{p\}$.
5:     Each $c$ computes $n_{i,c} \leftarrow \|U_c(:,i)\|_2^2$; all get $n_i \leftarrow \text{SECAGG}(\{n_{i,c}\})$.
6:     $Q_c(:,i) \leftarrow U_c(:,i)/\sqrt{n_i}$; set $R_{ii} \leftarrow \sqrt{n_i}$.
7:     **for each** $j \in \{i+1, \ldots, d\}$ **do**
8:         $r_{ij,c} \leftarrow Q_c(:,i)^\top U_c(:,j)$; all get $r_{ij} \leftarrow \text{SECAGG}(\{r_{ij,c}\})$.
9:         $R_{ij} \leftarrow r_{ij}$.
10:        $U_c(:,j) \leftarrow U_c(:,j) - Q_c(:,i)\, r_{ij}$.
11:        $d_j \leftarrow d_j - r_{ij}^2$.
12:     **end for**
13: **end for**
14: **Return to all sites:** $P$, $R$; **each site $c$ keeps:** $Q_c$ (so $Q = [Q_1, \cdots, Q_s]^\top$).

---

CPQR selects the column with the largest squared norm from the remaining unprocessed columns as the next pivot: $p = \arg\max_{j \in J} \|U(:,j)\|_2^2$, and swaps it into the $i$-th position. The $i$-th column is then normalized:

$$Q(:,i) = \frac{U(:,i)}{\|U(:,i)\|_2}, \qquad R_{ii} = \|U(:,i)\|_2. \tag{2}$$

All remaining columns $j > i$ are orthogonalized against $Q(:,i)$:

$$R_{ij} = Q(:,i)^\top U(:,j), \qquad U(:,j) \leftarrow U(:,j) - Q(:,i)R_{ij}. \tag{3}$$

However, centralized CPQR decomposition algorithm can not be applied directly due to the privacy constraints. To the best of our knowledge, CPQR in the context of FL remains largely unexplored.

In FL, data is distributed across multiple parties, i.e., $A = [A_1, \ldots, A_s]^\top$, and FedCPQR seeks a decomposition $[A_1, \ldots, A_s]^\top P \approx [Q_1, \ldots, Q_s]^\top R$, where $P$ is a permutation matrix reordering columns of $A$, $Q$ has orthonormal columns, and $R$ is upper triangular. Designing FedCPQR therefore requires addressing both distributed data partitioning and privacy preservation. To address these challenges, we propose to build on the modified Gram–Schmidt process to iteratively compute an orthogonal basis, while incorporating column pivoting in FL. The key observation is that both the pivoting and orthogonalization steps can be expressed as sums of local scalar quantities:

$$\|U(:,j)\|_2^2 = \sum_{c=1}^s \|U_c(:,j)\|_2^2, \qquad Q(:,i)^\top U(:,j) = \sum_{c=1}^s Q_c(:,i)^\top U_c(:,j). \tag{4}$$

These sums can be computed using secure aggregation such that no site exposes its raw data. After all iterations, the $R$ and permutation $P$ are known globally, while each site retains $Q_c$ that corresponds to its local data. Additionally, the squared norms of the remaining columns are updated after each deflation step to maintain accurate pivot selection:

$$d_j = \sum_{c=1}^s \|U_c(:,j)\|_2^2, \qquad d_j \leftarrow d_j - r_{ij}^2, \qquad r_{ij} = \sum_{c=1}^s Q_c(:,i)^\top U_c(:,j). \tag{5}$$

The main procedures of FedCPQR are summarized in Alg. 1. The $\text{SECAGG}(\cdot)$ represents the additive secure aggregation operation, which returns the sum across sites to all participants in clear text.

### 3.2.1 ANALYSES OF FEDCPQR

**Communication Cost** At each iteration $i$, FedCPQR relies on secure aggregation of scalars: (i) the column norm $n_i$; (ii) the projections $r_{ij}$ for all remaining columns $j > i$; Formally, the number of scalars aggregated at iteration $i$ is $\mathcal{O}(d^2)$, up to a constant factor, independent of the number of rows $n$. Hence, the communication cost scales quadratically with the number of features $d$ but is constant with respect to the number of samples $n$, making FedCPQR highly suitable for horizontal federated learning with large local datasets.

Table 1: Key differences between FedQR and FedCPQR.

|  | **FedQR** (Hartebrodt & Röttger, 2023) | **FedCPQR** (Proposed in this paper) |
|---|---|---|
| Rows of $A$ | Kept local | Kept local |
| Factor $Q_c$ | Kept local | Kept local |
| Shared factor | $R$ | $R, P$ |
| Also revealed | Global norms / inner products | Same plus pivot residual norms |
| Implied leakage | $A^\top A = R^\top R$ | $A^\top A = PR^\top RP^\top$ |
| Special-case risk | Raw data may leak if only 2 parties | Same |

**Computation Cost**   Each site performs standard local operations on its data $A_c \in \mathbb{R}^{n_c \times d}$: (i) computing local column norms $\|U_c(:,j)\|_2^2$ for $j \in J$, (ii) computing local projections $Q_c(:,i)^\top U_c(:,j)$, (iii) performing local deflation $U_c(:,j) \leftarrow U_c(:,j) - Q_c(:,i)r_{ij}$. At iteration $i$, the local computation cost is $\mathcal{O}(n_c(d-i))$ for norms and projections, plus $\mathcal{O}(n_c(d-i))$ for deflations, leading to a total per-site cost of $\mathcal{O}(n_c d^2)$ over all iterations. This matches the complexity of centralized CPQR (Quintana-Ortí et al., 1998) and scales linearly with local data size, making FedCPQR computationally practical in FL.

**Privacy Analysis**   FedCPQR communicates only secure sums of scalars: column norms, inner products, and residual norm updates. No raw rows, full columns, or local orthogonal factors $Q_c$ are shared. Concretely, the global outputs revealed are: (i) the permutation matrix $P$, exposing the relative column importance over iterations; (ii) the right triangular matrix $R$, hence $A^\top A = PR^\top RP^\top$ is revealed, (iii) the sequence of global residual norms $\{d_j\}$, used for pivot selection. Table 1 summarizes these differences between FedCPQR and FedQR.

**Correctness**   All pivot decisions and residual updates depend solely on sums of squared norms and inner products. These quantities decompose additively and can be securely aggregated, ensuring that FedCPQR reproduces the exact centralized CPQR on $A$. Formally, we state the following equivalence:

**Theorem 1.** *Assume exact secure aggregation of all required inner products and squared norms. The pivot choices and updates of FedCPQR are identical to those of centralized modified Gram–Schmidt based CPQR on the global matrix.*

### 3.3   GENERALIZED CUR DECOMPOSITION FOR FEATURE AND DATA SELECTION

Using the proposed FedCPQR as a sub-routine, we then propose the feature and data selection method based on FedGCUR algorithm. Specifically, FedGCUR uses one common column selection matrix across all parties: it first runs the FedCPQR (Alg. 1) to obtain the global permutation $P \in \mathbb{R}^{d \times d}$ and set

$$D = P(:,1\!:\!k) \in \mathbb{R}^{d \times k}, \qquad C = AD \in \mathbb{R}^{n \times k}, \qquad C_c = A_c D \in \mathbb{R}^{n_c \times k}.$$

Every silo then performs a local exact CPQR of $C_c^\top$ to select $r_c$ informative rows. $S_c \in \{0,1\}^{n_c \times r_c}$ is a row-selection matrix that picks those row indices (so $S_c^\top C_c \in \mathbb{R}^{r_c \times k}$ contains the selected rows of $C_c$), and define the local row submatrix $T_c := S_c^\top A_c \in \mathbb{R}^{r_c \times d}$. Finally, each silo computes its middle factor $M_c := C_c^\dagger A_c T_c^\dagger \in \mathbb{R}^{k \times r_c}$. The blockwise reconstruction stacks the local CURs:

$$\widehat{A}_{\mathrm{loc}} = \begin{bmatrix} C_1 M_1 T_1 \\ \vdots \\ C_s M_s T_s \end{bmatrix} = \mathrm{blkrow}_{c=1}^{s}\big(C_c M_c T_c\big).$$

The main steps of the proposed FedGCUR are summarized at Alg. 2. Note that, calculating the middle factor $M_c$ is optional. In the scenarios that only considers the feature and data selection, one can skip this step to reduce the computational cost.

#### 3.3.1   ANALYSIS OF FEDGCUR

**Computation, Communication and Privacy:**   Along with the cost of FedCPQR, each silo needs to perform an exact CPQR locally on the $C_c^\top \in \mathbb{R}^{k \times n_c}$ costs $\mathcal{O}(n_c k^2)$ flops to select $r_c$ rows.

---

**Algorithm 2** Data and Feature Selection by FedGCUR.

---

**Input:** $A = [A_1, \cdots, A_s]^\top \in \mathbb{R}^{n \times d}$; target rank $k$; per-silo quota $r_c \geq 1, c \in [s]$.
 1: Run FedCPQR (Alg. 1) on $A$ to obtain the $k$ pivot columns $D$ and set $C_c := A_c D \in \mathbb{R}^{n_c \times k}$.
 2: Each silo $c$ runs exact CPQR locally on $C_c^\top$ and outputs $r_c$ pivots $S_c$, and set $T_c := S_c^\top A_c$.
 3: (Optional) Compute $M_c \approx C_c^\dagger A_c T_c^\dagger$.
 4: **Return:** global $D$; each silo $c$ keeps $T_c, M_c, C_c$.

---

These costs exactly match the centralized pipeline applied to the horizontally concatenated $A$, up to negligible coordination overhead. Regarding the communication cost, no communication is needed for the local row selection and local $M_c$ formation. Therefore, FedGCUR does not introduce extra communication overhead, neither incurring any additional privacy concerns.

**Reconstruction Error Analysis**   To establish the bound of the reconstruction error, we first define the global column projector $P_C := CC^\dagger$ with $C = [C_1, \ldots, C_s]^\top$, and the block-diagonal row projector $P_R := \text{blkdiag}(P_{R,1}, \ldots, P_{R,s})$, where $P_{R,c} := T_c^\dagger T_c$ is the orthogonal projector onto the row space selected by CPQR on $C_c^\top$. Then, by the identity $M_c = C_c^\dagger A_c T_c^\dagger$ and standard manipulations,

$$A - \widehat{A}_{\text{loc}} = A - \text{blkrow}_{c=1}^s (C_c C_c^\dagger A_c T_c^\dagger T_c) = (I - P_C)A + P_C A(I - P_R). \tag{6}$$

We establish the following error bound. The proof is deferred to appendix.

**Theorem 2.** *Let $A \in \mathbb{R}^{n \times d}$ be partitioned by rows into $s$ parties, $A = [A_1, \ldots, A_s]^\top$, and $A_c \in \mathbb{R}^{n_c \times d}$. Let $\widehat{A}_{\text{loc}}$ be the FedGCUR reconstruction defined above, $k$ and $r_c$ be the number of selected columns and rows in party $c$, respectively. Then*

$$\|A - \widehat{A}_{\text{loc}}\| \leq \sqrt{1 + (d - k)\, 4^{k-1}}\; \sigma_{k+1}(A) + \max_{1 \leq c \leq s} \sqrt{1 + (n_c - r_c)\, 4^{r_c-1}}\; \sigma_{r_c+1}(C_c),$$

$$\tag{7}$$

$$\|A - \widehat{A}_{\text{loc}}\|_F \leq \sqrt{1 + (d - k)\, 4^{k-1}} \Big(\sum_{j>k} \sigma_j^2(A)\Big)^{1/2} + \Big(\sum_{c=1}^s \big[1 + (n_c - r_c)\, 4^{r_c-1}\big] \sum_{j>r_c} \sigma_j^2(C_c)\Big)^{1/2}.$$

$$\tag{8}$$

**Remark 1.**   Theorem 2 establishes an error bound of the reconstruction error of FedGCUR. The term $(I - P_C)A$ is a global column-selection residual driven by CPQR on the concatenated design; the term $P_C A(I - P_R)$ is a blockwise row-selection residual that aggregates local CPQR choices through the block-diagonal projector. It is worth emphasizing that FedGCUR is designed primarily for feature and data selection that preserves per-silo utility, rather than for optimizing a single global low-rank reconstruction. In light of this design goal, it is natural that its reconstruction error may not match that of classical CUR methods focused exclusively on global approximation, which aligns with our priority of per-client performance.

## 4 EXPERIMENT

### 4.1 EMPIRICAL SETTINGS

We conduct extensive experiments on a diverse set of datasets to evaluate the effectiveness and robustness of our proposed methods, FedCPQR and FedGCUR, across varying domains and data characteristics. Specifically, we use six datasets from the OpenML (Vanschoren et al., 2013) repository: mfeat-pixel, gina_prior2, devnagari-script, USPS, guillermo, and isolet, which differ in dimensionality and sample size. A summary of their characteristics is provided in Table 2.

To assess the correctness of FedCPQR, we adopt three performance metrics: pivot exact match, maximum principal angle, and projection distance, and compare our method against the QR decomposition with column pivoting implemented in SciPy (Virtanen et al., 2020) (i.e., `scipy.linalg.qr`). Specifically, let $AP_{fed} = Q_{fed}R_{fed}$ and $AP_{sci} = Q_{sci}R_{sci}$ denote the decompositions obtained by FedCPQR and SciPy, respectively. The metrics are defined as follows:

Table 2: Datasets Summary.

| Dataset | OpenML ID | # Train data | # Test data | # Features | # Classes |
|---|---|---|---|---|---|
| mfeat-pixel | 1022 | 1600 | 400 | 240 | 2 |
| gina_prior2 | 1041 | 2774 | 694 | 784 | 10 |
| devnagari-script | 40923 | 73600 | 18400 | 1024 | 46 |
| USPS | 41082 | 7438 | 1860 | 256 | 10 |
| guillermo | 41159 | 16000 | 4000 | 4296 | 2 |
| isolet | 43985 | 6237 | 1560 | 613 | 26 |

- *Pivot exact match* checks whether the set of selected columns by FedCPQR exactly matches the first $k$ pivot columns chosen by SciPy (i.e., comparing $P_{fed}$ and $P_{sci}$).
- *Maximum principal angle* measures the largest angular deviation between the subspaces spanned by $Q_{fed}$ and $Q_{sci}$, characterizing worst-case misalignment.
- *Projection distance* is computed as $\|Q_{fed}Q_{fed}^\top - Q_{sci}Q_{sci}^\top\|_F$, quantifying the aggregate deviation of $Q_{fed}$ from the reference subspace.

To evaluate the data and feature selection capability of FedGCUR, we use the selected subsets to train a global model and compare its accuracy against combinations of existing approaches: *Data selection:* Coreset (Sener & Savarese, 2018), Leverage Score (Larsen & Kolda, 2022), and Random. *Feature selection:* Maximum variance selection and Random. Note that data selection is performed locally at each party, with each selecting half of its local samples. For Leverage Score sampling, the selection scores are computed using the globally estimated $[Q_1, \ldots, Q_s]^\top$ from FedCPQR, which can be viewed as a degenerate variant of our method in the federated setting.

We empirically set the federated parties as 10. A three-layer neural network is employed as target model. FedAvg (McMahan et al., 2017) is used to aggregate the global model with a communication round of 10. The target decomposition column rank is selected from $k \in \{10, 50, 100\}$. We study 2 data split settings in the learning task experiment: uniform partitioning (i.i.d. split), Dirichlet distribution-based partitioning (non-i.i.d. split) Yurochkin et al. (2019) with concentration parameter $\alpha = 0.5$, where smaller $\alpha$ values correspond to higher data heterogeneity.

## 4.2 RESULTS

**Correctness of FedCPQR** We report the correctness results of FedCPQR in Table 3. The dataset is split i.i.d. for FL in this experiment. For all datasets and target ranks, the pivoted columns selected by FedCPQR exactly match those obtained by SciPy. The differences in maximum principal angle and projection distance are negligible, on the order of $10^{-14}$, and can be attributed to numerical precision loss. These findings confirm that FedCPQR achieves CPQR decomposition with accuracy comparable to the centralized algorithm, consistent with Theorem 1.

**Effectiveness of FedGCUR** We present the learning performance using the selected data and features in Table 4. The compared methods are abbreviated: the first word denotes the data selection method: Coreset, Leverage Score (Lever.), or Random (Rand.); while the second letter denotes the

Table 3: The decomposition results precision comparison between FedCPQR and SciPy.qr function.

| Dataset ID | Pivot Exactly Match | | | Max Principal Angle | | | Projection Dist. (Frobenius) | | |
|---|---|---|---|---|---|---|---|---|---|
| | k=10 | k=50 | k=100 | k=10 | k=50 | k=100 | k=10 | k=50 | k=100 |
| 1022 | True | True | True | 1.19e-15 | 5.97e-15 | 1.15e-14 | 2.27e-15 | 1.20e-14 | 2.52e-14 |
| 1041 | True | True | True | 1.02e-14 | 1.02e-14 | 1.02e-14 | 1.87e-15 | 1.35e-14 | 2.53e-14 |
| 40923 | True | True | True | 1.36e-14 | 2.23e-14 | 2.58e-14 | 6.77e-15 | 8.93e-15 | 1.13e-14 |
| 41082 | True | True | True | 1.13e-15 | 3.91e-15 | 9.50e-15 | 1.60e-15 | 9.70e-15 | 5.18e-14 |
| 41159 | True | True | True | 1.84e-15 | 2.07e-15 | 2.76e-15 | 1.37e-15 | 4.44e-15 | 8.32e-15 |
| 43985 | True | True | True | 1.53e-14 | 1.57e-14 | 1.86e-14 | 3.82e-15 | 5.96e-15 | 7.58e-15 |

Table 4: Task performance comparison of different data and feature selection methods on i.i.d. and non-i.i.d. split datasets. The mean and std. accuracy (%) are reported in the table. The best-performing method and all statistically comparable methods are highlighted in bold.

| ID | Rank | All data | FedCPQR | FedGCUR | Coreset.R | Coreset.V | Lever.R | Lever.V | Rand.R | Rand.V |
|---|---|---|---|---|---|---|---|---|---|---|
| | | | | Performance on IID Split Data | | | | | | |
| 1022 | 10 | 92.4±0.6 | **90.0±0.0** | 90.0±0.0 | 90.0±0.0 | 90.0±0.0 | 90.0±0.0 | 90.0±0.0 | 90.0±0.0 | 90.0±0.0 |
| | 50 | 92.4±0.6 | **90.0±0.0** | 90.0±0.0 | 90.0±0.0 | 90.0±0.0 | 90.0±0.0 | 90.0±0.0 | 90.0±0.0 | 90.0±0.0 |
| | 100 | 92.4±0.6 | **90.0±0.0** | **90.0±0.1** | **90.0±0.0** | **90.0±0.0** | **90.0±0.0** | **90.0±0.0** | **90.0±0.0** | **90.0±0.0** |
| 1041 | 10 | 58.8±1.3 | **11.1±1.6** | **10.4±2.0** | 10.5±0.0 | 10.5±0.0 | 10.5±0.0 | 10.6±0.1 | 10.2±0.0 | **11.1±0.0** |
| | 50 | 58.8±1.3 | **15.0±2.4** | **12.1±3.3** | 10.5±0.0 | 10.5±0.0 | 10.5±0.1 | 10.7±0.0 | 10.1±0.1 | 9.8±0.1 |
| | 100 | 58.8±1.3 | **15.4±3.4** | **17.5±2.4** | 11.5±0.6 | 11.5±0.6 | 14.8±0.8 | 10.1±0.1 | 15.3±0.6 | 12.2±0.6 |
| 40923 | 10 | 8.9±1.1 | **2.3±0.4** | **2.3±0.5** | **2.2±0.0** | **2.2±0.0** | **2.2±0.0** | **2.2±0.0** | **2.2±0.0** | **2.2±0.0** |
| | 50 | 8.9±1.1 | **2.6±0.5** | **2.2±0.5** | **2.2±0.0** | **2.2±0.0** | **2.2±0.0** | **2.2±0.0** | **2.2±0.0** | **2.2±0.0** |
| | 100 | 8.9±1.1 | **2.7±0.5** | **2.6±0.2** | 2.1±0.2 | 2.1±0.2 | **2.5±0.1** | 2.2±0.0 | 2.3±0.1 | 2.3±0.1 |
| 41082 | 10 | 60.0±0.8 | **26.4±3.8** | 20.9±7.0 | 17.2±0.7 | 17.2±0.7 | 19.1±0.5 | 16.3±0.6 | **28.5±1.4** | 24.1±1.0 |
| | 50 | 60.0±0.8 | 28.3±6.4 | **38.6±4.5** | 20.0±0.6 | 20.0±0.6 | 19.1±0.4 | 18.3±0.5 | 34.5±1.0 | 21.3±2.4 |
| | 100 | 60.0±0.8 | 42.2±4.7 | **46.9±2.8** | 25.7±1.2 | 25.7±1.2 | 37.2±1.1 | 29.9±0.6 | 37.0±1.0 | **46.2±0.8** |
| 41159 | 10 | 60.8±0.7 | **59.2±0.6** | 56.8±1.2 | **59.1±0.2** | **59.1±0.2** | 58.9±0.3 | **59.4±0.3** | 57.9±0.3 | 58.4±0.4 |
| | 50 | 60.8±0.7 | **59.0±0.6** | 58.8±0.5 | 58.2±0.5 | 58.2±0.5 | **58.7±0.2** | **58.6±0.3** | 53.6±1.3 | 56.2±0.7 |
| | 100 | 60.8±0.7 | 57.2±1.2 | **58.6±1.1** | 57.3±0.6 | 57.3±0.6 | **59.0±0.2** | 57.1±0.6 | 58.0±0.4 | 57.5±0.5 |
| 43985 | 10 | 78.7±1.2 | **15.8±2.7** | 12.7±1.6 | 14.3±0.6 | 14.3±0.6 | 12.2±0.6 | **16.9±0.8** | 14.8±1.0 | 15.4±0.8 |
| | 50 | 78.7±1.2 | **26.8±2.7** | **23.2±1.3** | 17.8±0.4 | 17.8±0.4 | 16.6±0.9 | 17.1±1.2 | 15.6±0.6 | 17.2±1.3 |
| | 100 | 78.7±1.2 | **35.7±1.8** | **37.9±2.0** | 24.4±0.8 | 24.4±0.8 | 21.2±0.5 | 21.5±0.9 | 21.4±1.0 | 24.8±2.0 |
| | | | | Performance on Non-IID Split Data | | | | | | |
| 1022 | 10 | 91.7±0.2 | **90.0±0.0** | 90.0±0.0 | 90.0±0.0 | 90.0±0.0 | 90.0±0.0 | 90.0±0.0 | 90.0±0.0 | 90.0±0.0 |
| | 50 | 91.7±0.2 | **90.0±0.0** | 90.0±0.0 | 90.0±0.0 | 90.0±0.0 | 90.0±0.0 | 90.0±0.0 | 90.0±0.0 | 90.0±0.0 |
| | 100 | 91.7±0.2 | **90.0±0.0** | 90.0±0.0 | 90.0±0.0 | 90.0±0.0 | 90.0±0.0 | 90.0±0.0 | 90.0±0.0 | 90.0±0.0 |
| 1041 | 10 | 58.8±1.4 | **11.0±1.6** | **10.3±2.1** | 10.5±0.0 | 10.5±0.0 | 10.5±0.0 | 10.6±0.1 | 10.2±0.0 | **10.9±0.3** |
| | 50 | 58.8±1.4 | **15.8±2.1** | **12.8±2.7** | 10.5±0.0 | 10.5±0.0 | 10.5±0.1 | 10.7±0.0 | 10.1±0.1 | 9.8±0.1 |
| | 100 | 58.8±1.4 | **16.3±2.2** | **17.6±1.4** | 11.0±0.4 | 11.0±0.4 | 14.5±0.9 | 10.1±0.1 | 15.1±0.4 | 12.0±0.4 |
| 40923 | 10 | 9.3±0.2 | **2.1±0.3** | **2.3±0.4** | **2.2±0.0** | **2.2±0.0** | **2.2±0.0** | **2.2±0.0** | **2.2±0.0** | **2.2±0.0** |
| | 50 | 9.3±0.2 | **2.5±0.5** | **2.3±0.5** | **2.2±0.0** | **2.2±0.0** | **2.2±0.0** | **2.2±0.0** | **2.2±0.0** | **2.2±0.0** |
| | 100 | 9.3±0.2 | 2.5±0.3 | **2.8±0.3** | 2.3±0.2 | 2.3±0.2 | 2.6±0.1 | 2.2±0.1 | 2.3±0.1 | 2.3±0.1 |
| 41082 | 10 | 59.8±0.7 | 26.8±4.1 | **31.9±6.3** | 17.7±1.0 | 17.7±1.0 | 20.0±0.5 | 17.6±1.1 | **28.7±0.7** | 24.5±1.0 |
| | 50 | 59.8±0.7 | 29.7±5.7 | **39.3±1.5** | 22.1±2.6 | 22.1±2.6 | 18.6±0.2 | 17.9±0.7 | 34.6±0.8 | 20.5±1.0 |
| | 100 | 59.8±0.7 | 41.8±4.8 | 41.5±3.0 | 26.1±1.7 | 26.1±1.7 | 35.7±0.8 | 31.2±1.2 | 36.9±0.5 | **46.1±0.6** |
| 41159 | 10 | 60.7±0.6 | 59.1±0.6 | **60.0±0.1** | 59.1±0.3 | 59.1±0.3 | 58.6±0.2 | 59.1±0.3 | 57.8±0.3 | 58.5±0.5 |
| | 50 | 60.7±0.6 | 59.0±0.5 | **60.0±0.0** | 58.1±0.5 | 58.1±0.5 | 58.3±0.5 | 58.4±0.3 | 53.6±1.3 | 55.9±0.9 |
| | 100 | 60.7±0.6 | 57.2±1.2 | **60.0±0.0** | 57.1±0.6 | 57.1±0.6 | 58.8±0.2 | 56.8±0.6 | 57.8±0.5 | 57.5±0.6 |
| 43985 | 10 | 73.3±2.5 | **14.6±1.9** | 11.7±2.7 | 12.8±1.4 | 12.8±1.4 | 11.1±1.6 | **14.9±1.7** | **14.1±0.7** | **14.2±1.0** |
| | 50 | 73.3±2.5 | **25.3±2.7** | **16.9±2.1** | 17.7±2.7 | 17.7±2.7 | 16.8±1.3 | 16.3±2.3 | 16.9±1.8 | 16.5±2.1 |
| | 100 | 73.3±2.5 | **32.7±2.9** | **25.9±0.8** | 24.4±1.2 | 24.4±1.2 | 21.1±0.8 | 21.0±1.1 | 20.2±1.5 | 21.7±1.4 |

feature selection method: Variance (V) or Random (R). For reference, we also report the performance of using all data, though this is not intended for direct comparison. In the case of FedCPQR, all data points are used along with sampled features, whereas all other methods select the same number of rows and columns. Each experiment is repeated 10 times with different random seeds, and average results are reported. The best-performing method and all statistically comparable methods are highlighted in bold. The significance test is obtained by conducting paired t-test of 0.05 significance level. Note that, when FedCPQR achieves the best performance, we instead highlight the second-best method and its comparable results, since FedCPQR uses all the data but the other methods only use a half.

The results show that FedCPQR and FedGCUR consistently achieve the best or near-best performance across most cases, demonstrating the effectiveness of the selected data and features. The performance trends are largely consistent between i.i.d. and non-i.i.d. settings. Notably, FedGCUR surpasses FedCPQR in several cases, highlighting the importance of jointly selecting both data and features. This is because harmful or noisy data may exist in the dataset; using such data not only increases communication and computation overhead, but can also degrade model performance. We

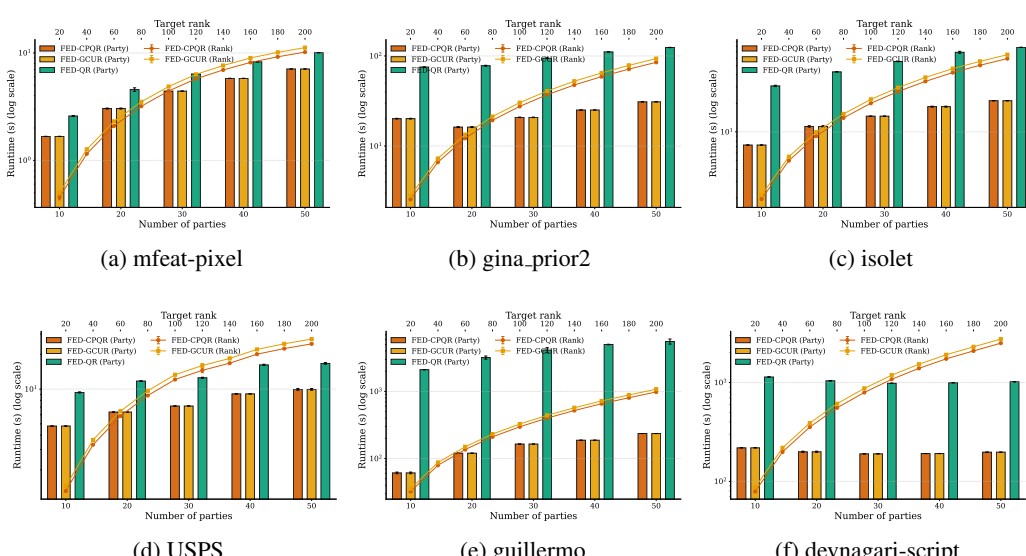

Figure 1: Log-scale running times (in seconds) of FedQR, FedCPQR, and FedGCUR with varying numbers of parties and ranks. Error bars indicating standard deviations of 5 runs.

also observe that FedGCUR benefits from higher target ranks, likely because row selection is performed on the already-selected columns, and a larger rank provides richer information for this step. Finally, the performance of baseline methods varies across datasets, implying the need for developing more specialized data and feature selection strategies for federated learning in the future. We note that the performance of all methods on dataset 40923 is relatively low. This is primarily due to the use of a limited number of communication rounds (10), which we fix across all datasets to ensure fairness in comparison. Since our focus in this experiment is on relative improvements, we retain the same training configuration throughout.

**Efficiency of FedCPQR and FedGCUR**  We report the log-scale running times (in seconds) of FedQR, FedCPQR, and FedGCUR under varying numbers of parties and target ranks in Figure 1. In the experiment varying the number of parties, the target rank is fixed at 50; in the experiment varying the target rank, the number of parties is fixed at 10. Each result is averaged over five runs, with error bars showing standard deviations. Since FedQR is independent of the target rank, it is excluded from the varying-rank experiment. The results show that the running times of FedCPQR and FedGCUR remain relatively low and do not increase substantially as either the number of parties or the target rank grows. Furthermore, both methods generally run faster than the FedQR baseline, highlighting their efficiency in handling high-dimensional data with relatively small target ranks in FL. In the varying-rank experiment, the running times of FedCPQR and FedGCUR are comparable, with FedGCUR being slightly more expensive. This suggests that the computational overhead of FedGCUR remains reasonable, making it practical for both data and feature selection.

## 5 CONCLUSION

In this paper, we proposed a unified framework for data and feature selection in federated learning based on generalized CUR decomposition. Specifically, we introduced FedCPQR, a federated column-pivoted QR algorithm that securely computes a global pivot order, and FedGCUR, which leverages these pivots to jointly perform feature and data selection. We provided communication, computation, and correctness analyses, showing that FedCPQR yields exactly the same decomposition as its centralized counterpart, and we established an upper bound on the reconstruction error of FedGCUR. Extensive experiments across datasets demonstrated that the proposed framework is both effective and efficient. Moreover, our work highlights the importance of jointly considering data and feature selection in federated learning. Future research may explore extensions of our framework to adaptive rank selection and domain-specific applications.

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

## A  LLM Usage Statement

The authors declare that Large Language Models (LLMs) were utilized in the preparation of this manuscript. Their use was limited to linguistic refinement, including polishing wording, improving sentence structure for clarity, and correcting spelling. Additionally, LLMs were employed to identify and correct minor errors in mathematical symbols and formulas. No part of the conceptualization, methodological design, data analysis, or substantive interpretation was generated by LLMs. The authors retain full responsibility for the content, accuracy, and originality of all scientific contributions presented in this paper.

## B  Reproducibility Statement

All experiments in this work were conducted using fixed random seeds to ensure reproducibility of the reported results. The experimental settings, including model configurations, training procedures, and evaluation protocols, were kept consistent across runs. To further support reproducibility, we will make the source code and implementation details publicly available at a later stage.

## C  Proof of Theorem 1

We restate Theorem 1 first, and then present the proof.

**Theorem 1.** *Assume exact secure aggregation of all required inner products and squared norms. The pivot choices and updates of FedCPQR are identical to those of centralized modified Gram–Schmidt based CPQR on the global matrix.*

*Proof.* All quantities used by modified Gram–Schmidt based CPQR can be expressed as sums over local contributions:

$$\|U(:,j)\|_2^2 = \sum_c \|U_c(:,j)\|_2^2, \qquad Q(:,i)^\top U(:,j) = \sum_c Q_c(:,i)^\top U_c(:,j).$$

Secure aggregation returns these exact global scalars. Hence, FedCPQR selects the same pivot at each iteration, sets the same $R_{ij}$, and updates the same residuals. The resulting $P$, $Q$, $R$, and derived projectors coincide with the centralized modified Gram–Schmidt based CPQR. □

## D  PROOF OF THEOREM 2

We first introduce the following lemma.

**Lemma 1** (Lemma 3.2 (Dong & Martinsson, 2021)). *Let $X \in \mathbb{R}^{\ell \times n}$ and run CPQR on $X$ to select $k$ pivot columns. Write the CPQR as*

$$X\Pi = Q \begin{bmatrix} T_1 & T_2 \end{bmatrix}, \tag{9}$$

*where $\Pi = [\Pi_c, \Pi_{cc}]$ is a permutation with $\Pi_c \in \mathbb{R}^{n \times k}$ extracting the first $k$ pivots, $Q \in \mathbb{R}^{\ell \times k}$ has orthonormal columns, and $T_1 \in \mathbb{R}^{k \times k}$ is nonsingular upper triangular. Define the right-acting oblique projector in the column-index space of $X$ by*

$$P_X = \Pi_c (X\Pi_c)^\dagger X = \Pi_c T_1^{-1} Q^\top X \in \mathbb{R}^{n \times n}. \tag{10}$$

*Then*

$$\|I_n - P_X\| \le \sqrt{1 + (n-k)\, 4^{\,k-1}}. \tag{11}$$

We restate theorem 2 here again for easier reading and present the proof.

**Theorem 2.** *Let $A \in \mathbb{R}^{n \times d}$ be partitioned by rows into $s$ parties, $A = \begin{bmatrix} A_1^\top, \ldots, A_s^\top \end{bmatrix}^\top$, and $A_c \in \mathbb{R}^{n_c \times d}$. Let $\widehat{A}_{\mathrm{loc}}$ be the FedGCUR reconstruction defined above, $k$ and $r_c$ be the number of selected columns and rows in party $c$, respectively. Then*

$$\|A - \widehat{A}_{\mathrm{loc}}\| \le \sqrt{1 + (d-k)\, 4^{k-1}}\, \sigma_{k+1}(A) + \max_{1 \le c \le s} \sqrt{1 + (n_c - r_c)\, 4^{r_c - 1}}\, \sigma_{r_c+1}(C_c),$$

$$\|A - \widehat{A}_{\mathrm{loc}}\|_F \le \sqrt{1 + (d-k)\, 4^{k-1}} \Big( \sum_{j>k} \sigma_j^2(A) \Big)^{1/2} + \Big( \sum_{c=1}^{s} \big[ 1 + (n_c - r_c)\, 4^{r_c - 1} \big] \sum_{j > r_c} \sigma_j^2(C_c) \Big)^{1/2}.$$

*Proof.* Since $P_C$ and $P_R$ are idempotent and $P_C$ is an orthogonal projector, we obtain

$$\|A - \widehat{A}_{\mathrm{loc}}\|_2 \le \|(I - P_C)A\|_2 + \|P_C A(I - P_R)\|_2 \le \|(I - P_C)A\|_2 + \|A(I - P_R)\|_2, \tag{12}$$

$$\|A - \widehat{A}_{\mathrm{loc}}\|_F^2 = \|(I - P_C)A\|_F^2 + \|P_C A(I - P_R)\|_F^2 \le \|(I - P_C)A\|_F^2 + \|A(I - P_R)\|_F^2. \tag{13}$$

The Frobenius equality above uses that the column spaces $\mathrm{range}(I - P_C)$ and $\mathrm{range}(P_C)$ are orthogonal.

Let $X \in \mathbb{R}^{n \times d}$ be the matrix on which CPQR is run to pick the $k$ column pivots. Write the CPQR as $X\Pi = Q \begin{bmatrix} R_1 & R_2 \end{bmatrix}$ where $\Pi = [\Pi_c, \Pi_{cc}]$ and $R_1 \in \mathbb{R}^{k \times k}$ is nonsingular. Define the right–acting oblique projector in the column–index space

$$P_X := \Pi_c (X\Pi_c)^\dagger X = \Pi_c R_1^{-1} Q^\top X \in \mathbb{R}^{d \times d}. \tag{14}$$

Then $X(I - P_X) = 0$ and

$$\|(I - P_C)A\|_\xi \le \|I - P_X\| \, \|A(I - P_k)\|_\xi, \qquad \xi \in \{2, F\}, \tag{15}$$

where $P_k$ is the orthogonal projector onto the $k$ selected coordinates. By Lemma 1, we have $\|I - P_X\| \le \sqrt{1 + (d-k)\, 4^{k-1}}$. By Eckart–Young–Mirsky theorem (Eckart & Young, 1936), we have $\|A(I - P_k)\|_2 = \sigma_{k+1}(A)$ and $\|A(I - P_k)\|_F = (\sum_{j>k} \sigma_j^2(A))^{1/2}$. This yields

$$\|(I - P_C)A\|_2 \le \sqrt{1 + (d-k)\, 4^{k-1}}\, \sigma_{k+1}(A), \tag{16}$$

$$\|(I - P_C)A\|_F \le \sqrt{1 + (d-k)\, 4^{k-1}} \Big( \sum_{j>k} \sigma_j^2(A) \Big)^{1/2}. \tag{17}$$

Fix a client $c$. Apply CPQR to $C_c^\top \in \mathbb{R}^{d \times n_c}$ to select $r_c$ rows, and let $P_{L,c} \in \mathbb{R}^{n_c \times n_c}$ be the associated left–acting oblique projector in the row–index space (constructed as in Lemma 1, with $(n, k) \leftarrow (n_c, r_c)$). Then

$$\|I_{n_c} - P_{L,c}\| \;\leq\; \sqrt{1 + (n_c - r_c)\, 4^{\, r_c - 1}}\,. \tag{18}$$

Let $P_{r_c}^{(c)}$ be the orthogonal projector onto the top $r_c$ right singular vectors of $C_c$. Again, by Eckart–Young–Mirsky theorem, we have

$$\|C_c(I - P_{r_c}^{(c)})\|_2 = \sigma_{r_c+1}(C_c), \qquad \|C_c(I - P_{r_c}^{(c)})\|_F = \Big( \sum_{j > r_c} \sigma_j^2(C_c) \Big)^{1/2}. \tag{19}$$

By a similar argument in the column analysis with transposed roles, we have

$$\|(I - P_{L,c})\, C_c\|_\xi \;\leq\; \|I - P_{L,c}\|\, \|C_c(I - P_{r_c}^{(c)})\|_\xi, \qquad \xi \in \{2, F\}. \tag{20}$$

Since $P_{R,c}$ is the orthogonal projector onto the selected row space, $C_c(I - P_{R,c}) = 0$, and hence for any $X_c$, $A_c(I - P_{R,c}) = (A_c - C_c)(I - P_{R,c})$, so that $\|A_c(I - P_{R,c})\|_\xi \leq \|(I - P_{L,c})\, C_c\|_\xi$. Combining the displays and using Eq. (18),

$$\|A_c(I - P_{R,c})\|_\xi \;\leq\; \sqrt{1 + (n_c - r_c)\, 4^{\, r_c - 1}} \times \begin{cases} \sigma_{r_c+1}(C_c), & \xi = 2, \\ \big( \sum_{j > r_c} \sigma_j^2(C_c) \big)^{1/2}, & \xi = F. \end{cases} \tag{21}$$

Stacking block rows yields

$$A(I - P_R) = \begin{bmatrix} A_1(I - P_{R,1}) \\ \vdots \\ A_s(I - P_{R,s}) \end{bmatrix}.$$

Hence

$$\|A(I - P_R)\|_2^2 \leq \sum_{c=1}^{s} \|A_c(I - P_{R,c})\|_2^2, \tag{22}$$

$$\|A(I - P_R)\|_F^2 = \sum_{c=1}^{s} \|A_c(I - P_{R,c})\|_F^2. \tag{23}$$

Insert Eq. (21) into Eq. (22)–Eq. (23) to obtain

$$\|A(I - P_R)\|_2 \leq \left( \sum_{c=1}^{s} \big[ 1 + (n_c - r_c)\, 4^{\, r_c - 1} \big]\, \sigma_{r_c+1}^2(C_c) \right)^{1/2}, \tag{24}$$

$$\|A(I - P_R)\|_F^2 \leq \sum_{c=1}^{s} \big[ 1 + (n_c - r_c)\, 4^{\, r_c - 1} \big] \sum_{j > r_c} \sigma_j^2(C_c). \tag{25}$$

Combine Eq. (12) with Eq. (16) and Eq. (24) to get Eq. (7); combine Eq. (13) with Eq. (17) and Eq. (25) to get Eq. (8). $\qquad \square$

# FEDERATED DATA AND FEATURE SELECTION BY GENERALIZED CUR DECOMPOSITION

**Anonymous authors**

## ABSTRACT

With the advance of federated learning (FL) in privacy-sensitive domains such as healthcare, finance, and mobile intelligence, the need for efficient and robust training becomes increasingly urgent. Communication bottlenecks, heterogeneous client distributions, and fairness requirements make it essential to select the "right" data and features for model training. Yet existing FL research often addresses feature selection and data selection separately, ignoring their interplay in real-world high-dimensional and noisy datasets, leading to suboptimal performance. In this paper, we propose a practical framework for joint data and feature selection based on generalized CUR decomposition. We introduce FedGCUR, which integrates a federated column-pivoted QR (FedCPQR) decomposition routine with per-silo row selection. FedCPQR exactly reproduces centralized CPQR pivoting in a horizontal FL setting by aggregating norms and inner products via secure addition, avoiding disclosure of raw rows or full columns while revealing the global permutation and triangular factor. FedGCUR leverages the resulting global column pivots to construct a shared feature set and heterogeneous, silo-specific row selections that respect local constraints. We show that FedCPQR matches the centralized CPQR decomposition, establish a reconstruction error bound for FedGCUR that captures the interplay between the global column projector and the block-diagonal row projector, and empirically validate that our framework achieves competitive or superior accuracy to existing data and feature selection baselines under both i.i.d. and non-i.i.d. splits.

## 1 INTRODUCTION

Federated learning (FL) has emerged as a key paradigm for collaborative model development across data silos and edge devices, enabling training without centralizing sensitive records (Zhang et al., 2021; Kairouz et al., 2021). By joint local learning and global model aggregation, FL leverages the knowledge of distributed clients in a privacy-preserving manner (Li et al., 2020). However, realizing these benefits in practice is non-trivial: statistical heterogeneity across silos (Ye et al., 2023), limited communication budgets (Niknam et al., 2020), and the risk of information leakage (Truex et al., 2019) during intermediate computations all complicate the training process. Addressing these issues often requires improving the efficiency and robustness of federated training, not only by designing better optimization protocols, but also by carefully selecting which data and features participate in the learning process.

Recently, numerous studies in federated learning have investigated data and feature selection (Fu et al., 2023; Hu et al., 2023; Wang et al., 2023). The main idea is to select a compact set of globally shared features to reduce client-side computation and uplink communication costs. Moreover, under participation or communication constraints, selecting more informative samples can further improve model prediction and distillation performance. Existing studies typically consider feature selection (Fu et al., 2023; Zhang et al., 2023; Hermo et al., 2024) and data selection (Liu et al., 2022; Li et al., 2021a; Rizk et al., 2022) as isolated subproblems rather than as interdependent decisions within a joint selection framework. As a concrete example, consider a high-dimensional dataset in healthcare with noisy, redundant features and heterogeneous label distributions across hospitals. Purely feature-centric criteria may favor globally high-variance biomarkers that neglect minority client populations, while purely data-centric criteria may prioritize samples that appear unique in the original space but become redundant or uninformative once projected into a more discriminative feature subspace. In

such settings, optimizing data and feature selection in isolation can waste the limited budget and lead to suboptimal or even contradictory decisions. Our goal is to explicitly capture this coupling in a practical joint selection framework based on generalized CUR decomposition (Garcia-Pedrajas et al., 2021; Dornaika, 2021; Fan et al., 2022).

One of the highly related techniques for data and feature selection is the matrix decomposition (Lu, 2022), which is fundamental to machine learning and data analysis (Caiafa et al., 2020; Sidiropoulos et al., 2017; Li et al., 2018). They are useful in dimension reduction, principal analysis, coreset discovery, etc. Unfortunately, most methods assume centralized access to the full matrix and thus do not directly apply in FL due to privacy and bandwidth constraints. Recent work has made preliminary progress on federated matrix decomposition (Chai et al., 2022; Hartebrodt & Röttger, 2023) by combining secure aggregation and data masking to avoid exposing raw data, but more expressive federated matrix decompositions remain underexplored.

Among decomposition-based selection methods, CUR is particularly appealing because it directly selects columns and rows of the data matrix Mahoney & Drineas (2009); Li et al. (2018). Given a matrix $A$, CUR seeks factors $A \approx CMT$, where $C$ collects a subset of columns (features), $T$ collects a subset of rows (examples), and $M$ is a small linking matrix. In FL, however, CUR must satisfy federated constraints that do not arise centrally: (i) select a shared set of features so that a joint model can be trained and aggregated across silos; (ii) reveal only privacy-preserving aggregated statistics. These requirements make the federated CUR problem challenging, because rank-revealing selection typically needs global coordination; and heterogeneous silos may prefer different pivots; and naive protocols either leak information or incur prohibitive costs.

In this paper, we bridge this gap by formulating federated data and feature selection as a federated generalized CUR (FedGCUR) decomposition problem. To the best of our knowledge, this is the first work that studies joint data and feature selection in a cross-silo FL setting through a generalized CUR lens. Our practical framework FedGCUR has two tightly coupled components. First, we develop FedCPQR, a federated column-pivoted QR routine that, to the best of our knowledge, is the first protocol to exactly reproduce centralized CPQR pivoting in horizontal FL. It does so by aggregating local squared norms and inner products via secure addition while tracking pivot residuals, thereby revealing a global permutation and triangular factor but never exposing raw rows or full columns, in line with the information revealed by FedQR (Hartebrodt & Röttger, 2023). Second, we build FedGCUR on top of FedCPQR: a single global column index set, shared across silos, is used to define a common feature subspace, while each silo independently performs local row selection to respect heterogeneous data and budget constraints. We provide communication and computation analyses, show that FedCPQR produces the same decomposition as centralized modified Gram–Schmidt CPQR under exact aggregation, and derive a reconstruction error bound for FedGCUR that explicitly captures the interaction between the global column projector and the block-diagonal row projector. Extensive experiments under both i.i.d. and Dirichlet non-i.i.d. splits demonstrate that FedGCUR achieves competitive or superior utility relative to strong data and feature selection baselines.

## 2 RELATED WORKS

**Federated data selection and feature selection**   Recent work explores privacy-preserving data and feature selection in federated settings (Fu et al., 2023; Zhang et al., 2023; Li et al., 2021a; Rizk et al., 2022). For example, Hu et al. (2023) propose a heuristic federated feature selection scheme where each client performs local subset selection and shares only the selected feature indices and their local accuracy, after re-evaluating others' subsets, and the aggregator selects a global subset via an aggregation rule. Wang et al. (2023) propose FAST, a federated learning framework that jointly adapts class-wise data sampling and the number of local iterations to counter Non-IID data and device heterogeneity, with an online MAB-based controller and a proved convergence bound. Hermo et al. (2024) propose Fed-mRMR, a lossless federated adaptation of the classic mRMR filter that replaces raw-data sharing with an occurrences matrix built from local bitmaps and merged across clients, preserving the exact mRMR ranking under non-IID, cross-silo FL. Most existing methods focus on either data selection or feature selection in isolation, which can be suboptimal when both are required, as they ignore the coupling between instances and features.

**Federated matrix decomposition** Matrix decomposition in federated learning receives less attention. Strong privacy constraints preclude sharing matrix statistics, making standard decompositions difficult to adapt. Hartebrodt & Röttger (2023) studied federated QR decomposition. A modified Gram-Schmidt based QR decomposition method is proposed for federated learning settings. This method approximates $[A_1, \ldots, A_s]^\top \approx [Q_1, \ldots, Q_s]^\top R$, where each client $c$ can only access its corresponding local matrix $Q_c$. The right triangular matrix $R$ is shared globally. Specifically, each site locally maintains its rows and iteratively performs Gram-Schmidt steps, with the only cross-site communication being the sums of inner products and norms. An additive aggregation protocol is employed to securely aggregate the necessary scalars. Chai et al. (2022) investigate FedSVD for vertical FL. A trusted authority generates a unitary random mask, clients upload masked data, and the server computes an SVD on the masked matrix under secure aggregation. Each client receives the right singular vectors corresponding to its features, while the left singular vectors and singular values are shared globally. Tensor decomposition in FL (Gao et al., 2021; Ouyang et al., 2024) has also been explored, but it lies beyond the scope of this paper. In addition, there is a growing line of work on federated matrix factorizations and component-wise decompositions, such as federated NMF (Si et al., 2022; Dalleiger & Gionis, 2025) and related low-rank models (Li et al., 2021b; Dadras et al., 2024), that aim to reconstruct or approximate the full data matrix (or tensors) under privacy constraints. These methods typically focus on learning latent factors for representation learning or collaborative filtering, rather than selecting physical rows and columns. By contrast, our work targets joint data and feature selection under communication and participation budgets, where the goal is to identify informative subsets of actual samples and features that can be reused across downstream tasks. This difference in objective and output structure leads to different algorithmic requirements and makes generalized CUR a natural tool in our setting.

## 3 METHODOLOGY

### 3.1 PRELIMINARIES

**Notations** In this paper, matrices or sets are denoted by capital letters, vectors by bold lowercase letters, and scalars by lowercase letters. We use superscript † to represent the pseudo-inverse of a matrix, and $\sigma_j$ represents the $j$-th largest singular value. An $m \times m$ identity matrix is denoted by $I_m$, and we omit the subscript when the context is clear. Denote by $\| \cdot \|_2$ the spectral norm for matrix and $\ell_2$ norm for vector, and $\| \cdot \|_F$ the Frobenius norm. We define $[n] = \{1, 2, \ldots, n\}$.

This work focuses primarily on cross-silo horizontal FL scenarios. Specifically, given a data matrix $A \in \mathbb{R}^{n \times d}$, where each row corresponds to data from $s$ different sites, we have $A = [A_1, \ldots, A_s]^\top$, with $A_c \in \mathbb{R}^{n_c \times d}$, and $\sum_{c=1}^{s} n_c = n$. Here, $c$ represents the client index. We assume that each row of $A$ is randomly sampled from an unknown distribution, though the split function across clients is arbitrary. The target vector is denoted as $\boldsymbol{b} \in \mathbb{R}^{n \times 1}$. Throughout, $k$ is the target column rank, $j$ represents the column index, and $r_c$ is the number of selected data point in silo $c$, with $r = \sum_{c=1}^{s} r_c$ and $r_c \leq n_c$.

**Secure Aggregation Protocol** Following Hartebrodt & Röttger (2023), we use an additive secure aggregation protocol (Cramer et al., 2015) which, for a set of scalar values $\{x_c\}_{c=1}^{s}$, returns the sum $\sum_{c=1}^{s} x_c$ to all participants without revealing the individual $x_c$ values. Concretely, each client masks its local scalar with random values drawn modulo a public large prime, and the masks cancel when summed, so that only the aggregate is revealed.

**Threat model** Throughout this work, we adopt an honest-but-curious (passive) adversary model. All parties follow the prescribed protocol but may attempt to infer additional information from the messages they observe. We assume at least three non-colluding silos participate in the secure aggregation rounds, as is standard in cross-silo FL deployments. Under this model, our protocols avoid disclosure of raw rows or full columns from any client: only aggregated scalar statistics (sums of squared norms and inner products), the global permutation matrix, and the upper-triangular factor are revealed. We do not claim differential privacy guarantees, and we explicitly acknowledge that revealing $P$, $R$, and residual norms implies access to $A^\top A$ up to permutation. Note that, this information leakage is analogous to FedQR (Hartebrodt & Röttger, 2023)

**CUR and Generalized CUR Decomposition** CUR decomposition seeks a subset of $k$ rows and columns from $A \in \mathbb{R}^{n \times d}$ such that $A \approx CMT = ADMS^\top A$, where $C \in \mathbb{R}^{n \times k}$ and $T \in \mathbb{R}^{k \times d}$ are sub-matrices of the columns and rows of $A$, respectively. Here, $D \in \mathbb{R}^{d \times k}$ and $S \in \mathbb{R}^{n \times k}$ are *index selection matrices*, constructed from columns of the identity matrix $I$ with appropriate permutations.

Generalized CUR decomposition is usually defined on a pair of matrices (Gidisu & Hochstenbach, 2022; Cao et al., 2024). Here, we generalize its definition to $s$ matrices: $A_1, \ldots, A_s$. The formal definition is as follows:

**Definition 1** (Generalized CUR Decomposition for Multiple Matrices). *Let $s \in \mathbb{N}, s > 1$, and $c \in [s]$, $\{A_c\}_{c=1}^s$ be a collection of matrices where $A_c \in \mathbb{R}^{n_c \times d}$, $n_c \geq d$. Given the number of sampled rows $r_c$ for $A_c$ (with $r_c \leq n_c$) and the number of sampled columns $k$ (with $k \leq d$), the generalized CUR decomposition of $\{A_c\}_{c=1}^s$ seeks approximations of the form*

$$A_c \approx C_c M_c T_c = A_c D M_c S_c^\top A_c, \tag{1}$$

*where, $S_c \in \mathbb{R}^{n_c \times r_c}$ and $D \in \mathbb{R}^{d \times k}$ are index selection matrices. When $s = 2$, the decomposition problem coincides with Gidisu & Hochstenbach (2022).*

It is important to note that all $\{A_c\}_{c=1}^s$ share the same index selection matrix $D$ in the generalized CUR decomposition.

## 3.2 FEDERATED CPQR

Considering federated data and feature selection, it is necessary to perform feature selection globally without exposing the raw data held by each client. This task can be formulated as a federated column-pivoted QR decomposition problem. The primary difference between standard QR (Watkins, 1982) and column-pivoted QR (CPQR) (Quintana-Ortí et al., 1998) lies in the column selection strategy. Specifically, let $U = [U_1, \ldots, U_s]^\top$ be the working matrix. At iteration $i$, CPQR selects the column with the largest squared norm from the remaining unprocessed columns as the next pivot: $p = \arg\max_{j \in J} \|U(:,j)\|_2^2$, and swaps it into the $i$-th position. The $i$-th column is then normalized:

$$Q(:,i) = \frac{U(:,i)}{\|U(:,i)\|_2}, \qquad R_{ii} = \|U(:,i)\|_2. \tag{2}$$

All remaining columns $j > i$ are orthogonalized against $Q(:,i)$:

$$R_{ij} = Q(:,i)^\top U(:,j), \qquad U(:,j) \leftarrow U(:,j) - Q(:,i)R_{ij}. \tag{3}$$

However, centralized CPQR decomposition algorithm can not be applied directly due to the privacy constraints. While Hartebrodt & Röttger (2023) study federated QR without pivoting, CPQR with exact centralized pivot reproduction in a horizontal FL setting remains largely unexplored.

In FL, data is distributed across multiple parties, i.e., $A = [A_1, \ldots, A_s]^\top$, and FedCPQR seeks a decomposition $[A_1, \ldots, A_s]^\top P \approx [Q_1, \ldots, Q_s]^\top R$, where $P$ is a permutation matrix reordering columns of $A$, $Q$ has orthonormal columns, and $R$ is upper triangular. Designing FedCPQR therefore requires addressing both distributed data partitioning and privacy preservation. Compared to FedQR (Hartebrodt & Röttger, 2023), which exposes only the triangular factor and uses non-pivoted Gram–Schmidt, FedCPQR must additionally track and share the evolving permutation and residual norms to support rank-revealing column selection, while still communicating only aggregated scalars. To address these challenges, we propose to build on the modified Gram–Schmidt process to iteratively compute an orthogonal basis, while incorporating column pivoting in FL. The key observation is that both the pivoting and orthogonalization steps can be expressed as sums of local scalar quantities:

$$\|U(:,j)\|_2^2 = \sum_{c=1}^s \|U_c(:,j)\|_2^2, \qquad Q(:,i)^\top U(:,j) = \sum_{c=1}^s Q_c(:,i)^\top U_c(:,j). \tag{4}$$

These sums can be computed using secure aggregation such that no site exposes its raw data. After all iterations, the $R$ and permutation $P$ are known globally, while each site retains $Q_c$ that corresponds to its local data. Additionally, the squared norms of the remaining columns are updated after each deflation step to maintain accurate pivot selection:

$$d_j = \sum_{c=1}^s \|U_c(:,j)\|_2^2, \qquad d_j \leftarrow d_j - r_{ij}^2, \qquad r_{ij} = \sum_{c=1}^s Q_c(:,i)^\top U_c(:,j). \tag{5}$$

---

**Algorithm 1** FedCPQR

---

**Input:** Each site $c \in [s]$ holds $A_c \in \mathbb{R}^{n_c \times d}$.
1: **Initialize:** $J \leftarrow [d]$, $P \leftarrow I_d$, $R \leftarrow 0_{d \times d}$. $U_c \leftarrow A_c$, for $c \in [s]$.
2: For each $j \in J$, $c \in [s]$: compute $d_{j,c} \leftarrow \|U_c(:,j)\|_2^2$; $d_j \leftarrow \text{SECAGG}(\{d_{j,c}\}_{c=1}^s)$.
3: **for** $i = 1$ **to** $d$ **do**
4:     $p \leftarrow \arg\max_{j \in J} d_j$. Swap columns $i \leftrightarrow p$ in $P$, in each local $U_c$, and in the first $i-1$ rows of $R$; update $J \leftarrow J \setminus \{p\}$.
5:     Each $c$ computes $n_{i,c} \leftarrow \|U_c(:,i)\|_2^2$; all get $n_i \leftarrow \text{SECAGG}(\{n_{i,c}\})$.
6:     $Q_c(:,i) \leftarrow U_c(:,i)/\sqrt{n_i}$; set $R_{ii} \leftarrow \sqrt{n_i}$.
7:     **for each** $j \in \{i+1, \ldots, d\}$ **do**
8:         $r_{ij,c} \leftarrow Q_c(:,i)^\top U_c(:,j)$; all get $r_{ij} \leftarrow \text{SECAGG}(\{r_{ij,c}\})$.
9:         $R_{ij} \leftarrow r_{ij}$.
10:        $U_c(:,j) \leftarrow U_c(:,j) - Q_c(:,i)\, r_{ij}$.
11:        $d_j \leftarrow d_j - r_{ij}^2$.
12:     **end for**
13: **end for**
14: **Return to all sites:** $P$, $R$; **each site $c$ keeps:** $Q_c$ (so $Q = [Q_1, \cdots, Q_s]^\top$).

---

The main procedures of FedCPQR are summarized in Alg. 1. The $\text{SECAGG}(\cdot)$ represents the additive secure aggregation operation, which returns the sum across sites to all participants in clear text.

### 3.2.1 ANALYSES OF FEDCPQR

**Communication Cost** At each iteration $i$, FedCPQR relies on secure aggregation of scalars: (i) the column norm $n_i$; (ii) the projections $r_{ij}$ for all remaining columns $j > i$; Formally, the number of scalars aggregated at iteration $i$ is $\mathcal{O}(d^2)$, up to a constant factor, independent of the number of rows $n$. In particular, our analytical communication complexity aligns with the scalar-count expressions and is most advantageous in regimes with large sample sizes $n$ and moderate feature dimension $d$, which are common in cross-silo applications such as healthcare.

**Computation Cost** Each site performs standard local operations on its data $A_c \in \mathbb{R}^{n_c \times d}$: (i) computing local column norms $\|U_c(:,j)\|_2^2$ for $j \in J$, (ii) computing local projections $Q_c(:,i)^\top U_c(:,j)$, (iii) performing local deflation $U_c(:,j) \leftarrow U_c(:,j) - Q_c(:,i)r_{ij}$. At iteration $i$, the local computation cost is $\mathcal{O}(n_c(d-i))$ for norms and projections, plus $\mathcal{O}(n_c(d-i))$ for deflations, leading to a total per-site cost of $\mathcal{O}(n_c d^2)$ over all iterations. This matches the complexity of centralized CPQR (Quintana-Ortí et al., 1998) and scales linearly with local data size, making FedCPQR computationally practical in FL.

**Privacy Analysis** FedCPQR communicates only secure sums of scalars: column norms, inner products, and residual norm updates; so that no client ever sends raw rows, complete columns, or its local orthogonal factor $Q_c$. The globally revealed quantities are: (i) the permutation matrix $P$, which reflects the relative ordering of column pivots; (ii) the upper-triangular matrix $R$, from which $A^\top A = PR^\top R P^\top$ can be inferred; and (iii) the sequence of global residual norms $\{d_j\}$ used for pivot selection. As summarized in Table 3, this implied leakage is analogous to FedQR (Hartebrodt & Röttger, 2023), with the additional exposure of $P$ and residual norms needed for rank-revealing pivoting. Under our honest-but-curious threat model with at least three non-colluding silos, this design avoids disclosure of any participant's raw data while enabling exact CPQR pivot reproduction.

**Correctness** All pivot decisions and residual updates depend solely on sums of squared norms and inner products. These quantities decompose additively and can be securely aggregated, ensuring that FedCPQR reproduces the exact centralized CPQR on $A$. Formally, we state the following equivalence:

**Theorem 1.** *Assume exact secure aggregation of all required inner products and squared norms. The pivot choices and updates of FedCPQR are identical to those of centralized modified Gram–Schmidt based CPQR on the global matrix.*

---

**Algorithm 2** Data and Feature Selection by FedGCUR.

---

**Input:** $A = [A_1, \cdots, A_s]^\top \in \mathbb{R}^{n \times d}$; target rank $k$; per-silo quota $r_c \geq 1, c \in [s]$.
1: Run FedCPQR (Alg. 1) on $A$ to obtain the $k$ pivot columns $D$ and set $C_c := A_c D \in \mathbb{R}^{n_c \times k}$.
2: Each silo $c$ runs exact CPQR locally on $C_c^\top$ and outputs $r_c$ pivots $S_c$, and set $T_c := S_c^\top A_c$.
3: (Optional) Compute $M_c \approx C_c^\dagger A_c T_c^\dagger$.
4: **Return:** global $D$; each silo $c$ keeps $T_c, M_c, C_c$.

---

In Section 4, we empirically confirm this equivalence by comparing FedCPQR against the `scipy.linalg.qr` implementation with column pivoting, observing exact pivot-order matches and negligible numerical differences in the resulting subspaces.

### 3.3 GENERALIZED CUR DECOMPOSITION FOR FEATURE AND DATA SELECTION

Using the proposed FedCPQR as a sub-routine, we then propose the feature and data selection method based on FedGCUR algorithm. Specifically, FedGCUR uses one common column selection matrix across all parties: it first runs the FedCPQR (Alg. 1) to obtain the global permutation $P \in \mathbb{R}^{d \times d}$ and set

$$D = P(:, 1{:}k) \in \mathbb{R}^{d \times k}, \qquad C = AD \in \mathbb{R}^{n \times k}, \qquad C_c = A_c D \in \mathbb{R}^{n_c \times k}.$$

Every silo then performs a local exact CPQR of $C_c^\top$ to select $r_c$ informative rows. $S_c \in \{0,1\}^{n_c \times r_c}$ is a row-selection matrix that picks those row indices (so $S_c^\top C_c \in \mathbb{R}^{r_c \times k}$ contains the selected rows of $C_c$), and define the local row submatrix $T_c := S_c^\top A_c \in \mathbb{R}^{r_c \times d}$. Finally, each silo computes its middle factor $M_c := C_c^\dagger A_c T_c^\dagger \in \mathbb{R}^{k \times r_c}$. The blockwise reconstruction stacks the local CURs:

$$\widehat{A}_{\text{loc}} = \begin{bmatrix} C_1 M_1 T_1 \\ \vdots \\ C_s M_s T_s \end{bmatrix} = \text{blkrow}_{c=1}^s \big( C_c M_c T_c \big).$$

The main steps of the proposed FedGCUR are summarized at Alg. 2. Note that, calculating the middle factor $M_c$ is optional. In the scenarios that only considers the feature and data selection, one can skip this step to reduce the computational cost.

#### 3.3.1 ANALYSIS OF FEDGCUR

**Computation, Communication and Privacy:** Along with the cost of FedCPQR, each silo needs to perform an exact CPQR locally on the $C_c^\top \in \mathbb{R}^{k \times n_c}$ costs $\mathcal{O}(n_c k^2)$ flops to select $r_c$ rows. These costs exactly match the centralized pipeline applied to the horizontally concatenated $A$, up to negligible coordination overhead. Regarding the communication cost, no communication is needed for the local row selection and local $M_c$ formation. Therefore, FedGCUR does not introduce extra communication overhead, neither incurring any additional privacy concerns.

If all data were centralized, applying generalized CUR directly to $A$ would yield a specific GCUR instance with a single column index set and row selections over the stacked matrix. FedGCUR recovers this structure in a distributed manner: the global column index set is computed without raw data access, and row selection is performed locally under per-silo budgets and constraints. The difficulty therefore lies not in generalizing the algebraic definition of GCUR, which follows Gidisu & Hochstenbach (2022); Cao et al. (2024) but in instantiating it under federated communication and privacy constraints.

**Reconstruction Error Analysis** To establish the bound of the reconstruction error, we first define the global column projector $P_C := CC^\dagger$ with $C = [C_1, \ldots, C_s]^\top$, and the block-diagonal row projector $P_R := \text{blkdiag}(P_{R,1}, \ldots, P_{R,s})$, where $P_{R,c} := T_c^\dagger T_c$ is the orthogonal projector onto the row space selected by CPQR on $C_c^\top$. Then, by the identity $M_c = C_c^\dagger A_c T_c^\dagger$ and standard manipulations,

$$A - \widehat{A}_{\text{loc}} = A - \text{blkrow}_{c=1}^s(C_c C_c^\dagger A_c T_c^\dagger T_c) = (I - P_C)A + P_C A(I - P_R). \tag{6}$$

We establish the following error bound. The proof is deferred to appendix.

**Theorem 2.** *Let $A \in \mathbb{R}^{n \times d}$ be partitioned by rows into $s$ parties, $A = [A_1, \ldots, A_s]^\top$, and $A_c \in \mathbb{R}^{n_c \times d}$. Let $\widehat{A}_{\mathrm{loc}}$ be the FedGCUR reconstruction defined above, $k$ and $r_c$ be the number of selected columns and rows in party $c$, respectively. Then*

$$\|A - \widehat{A}_{\mathrm{loc}}\| \leq \sqrt{1 + (d - k)\, 4^{k-1}}\ \sigma_{k+1}(A)\ +\ \max_{1 \leq c \leq s}\ \sqrt{1 + (n_c - r_c)\, 4^{r_c - 1}}\ \sigma_{r_c+1}(C_c),$$

(7)

$$\|A - \widehat{A}_{\mathrm{loc}}\|_F \leq \sqrt{1 + (d - k)\, 4^{k-1}} \Big(\sum_{j>k} \sigma_j^2(A)\Big)^{1/2} + \Big(\sum_{c=1}^{s} \big[1 + (n_c - r_c)\, 4^{r_c - 1}\big] \sum_{j>r_c} \sigma_j^2(C_c)\Big)^{1/2}.$$

(8)

**Remark 1.** Theorem 2 establishes an error bound of the reconstruction error of FedGCUR. The term $(I - P_C)A$ is a global column-selection residual driven by CPQR on the concatenated design; the term $P_C A(I - P_R)$ is a blockwise row-selection residual that aggregates local CPQR choices through the block-diagonal projector. It is worth emphasizing that FedGCUR is designed primarily for feature and data selection that preserves per-silo utility, rather than for optimizing a single global low-rank reconstruction. In light of this design goal, it is natural that its reconstruction error may not match that of classical CUR methods focused exclusively on global approximation, which aligns with our priority of per-client performance.

## 4 EXPERIMENT

### 4.1 EMPIRICAL SETTINGS

We conduct extensive experiments on a diverse set of datasets to evaluate the effectiveness and robustness of our proposed methods, FedCPQR and FedGCUR, across varying domains and data characteristics. Specifically, we use six datasets from the OpenML (Vanschoren et al., 2013) repository: mfeat-pixel, gina_prior2, devnagari-script, USPS, guillermo, and isolet, which differ in dimensionality and sample size. A summary of their characteristics is provided in Table 4.

To assess the correctness of FedCPQR, we adopt three performance metrics: pivot exact match, maximum principal angle, and projection distance, and compare our method against the QR decomposition with column pivoting implemented in SciPy (Virtanen et al., 2020) (i.e., `scipy.linalg.qr`). Specifically, let $AP_{fed} = Q_{fed}R_{fed}$ and $AP_{sci} = Q_{sci}R_{sci}$ denote the decompositions obtained by FedCPQR and SciPy, respectively. The metrics are defined as follows:

- *Pivot exact match* checks whether the set of selected columns by FedCPQR exactly matches the first $k$ pivot columns chosen by SciPy (i.e., comparing $P_{fed}$ and $P_{sci}$).
- *Maximum principal angle* measures the largest angular deviation between the subspaces spanned by $Q_{fed}$ and $Q_{sci}$, characterizing worst-case misalignment.
- *Projection distance* is computed as $\|Q_{fed}Q_{fed}^\top - Q_{sci}Q_{sci}^\top\|_F$, quantifying the aggregate deviation of $Q_{fed}$ from the reference subspace.

To evaluate the data and feature selection capability of FedGCUR, we use the selected subsets to train a global model and compare its accuracy against combinations of existing approaches: *Data selection:* Coreset (Sener & Savarese, 2018), Leverage Score (Larsen & Kolda, 2022), and Random. *Feature selection:* Maximum variance selection and Random. Note that data selection is performed locally at each party, with each selecting half of its local samples. For Leverage Score sampling, the selection scores are computed using the globally estimated $[Q_1, \ldots, Q_s]^\top$ from FedCPQR, which can be viewed as a degenerate variant of our method in the federated setting.

We empirically set the federated parties as 10. A three-layer neural network is employed as target model. FedAvg (McMahan et al., 2017) is used to aggregate the global model with a communication round of 10. The target decomposition column rank is selected from $k \in \{10, 50, 100\}$. We study 2 data split settings in the learning task experiment: uniform partitioning (i.i.d. split), Dirichlet distribution-based partitioning (non-i.i.d. split) Yurochkin et al. (2019) with concentration parameter $\alpha = 0.5$, where smaller $\alpha$ values correspond to higher data heterogeneity.

## 4.2 RESULTS

**Correctness of FedCPQR**   We report the correctness results of FedCPQR in Table 1. The dataset is split i.i.d. for FL in this experiment. For all datasets and target ranks, the pivoted columns selected by FedCPQR exactly match those obtained by SciPy. The differences in maximum principal angle and projection distance are negligible, on the order of $10^{-14}$, and can be attributed to numerical precision loss. These findings confirm that FedCPQR achieves CPQR decomposition with accuracy comparable to the centralized algorithm, consistent with Theorem 1.

**Effectiveness of FedGCUR**   To evaluate the data and feature selection capability of FedGCUR, we use the selected subsets to train a global model and compare its accuracy against combinations of existing approaches: *Data selection:* Coreset (Sener & Savarese, 2018), Leverage Score (Larsen & Kolda, 2022), and Random. *Feature selection:* Maximum variance selection and Random. Note that data selection is performed locally at each party, with each selecting half of its local samples. For Leverage Score sampling, the selection scores are computed using the globally estimated $[Q_1, \ldots, Q_s]^\top$ from FedCPQR, which can be viewed as a degenerate variant of our method in the federated setting. Consequently, the Leverage Score baseline effectively serves as an ablation that replaces our pivot-based row selection with a non-pivoted FedQR-style approach using leverage scores computed from the (federated) QR subspace. This isolates the incremental benefit of explicit pivoting in FedCPQR/FedGCUR over a closely related non-pivoted alternative.

We present the learning performance using the selected data and features in Table 2. The compared methods are abbreviated: the first word denotes the data selection method: Coreset, Leverage Score (Lever.), or Random (Rand.); while the second letter denotes the feature selection method: Variance (V) or Random (R). Note that, we treat the models trained on all available data ("All Data") and those using FedCPQR with full sample access as oracle or upper-bound baselines: they characterize the maximum achievable performance when the full dataset is available to the selection mechanism, but do not correspond to budget-constrained or communication-limited deployment regimes. Our comparisons of FedGCUR and other selection methods should therefore be interpreted relative to these upper bounds rather than as direct efficiency competitors. All the compared methods select the same number of rows and columns. Each experiment is repeated 10 times with different random seeds, and average results are reported. The best-performing method and all statistically comparable methods are highlighted in bold. The significance test is obtained by conducting paired t-test of 0.05 significance level. Note that, when FedCPQR achieves the best performance, we instead highlight the second-best method and its comparable results, since FedCPQR uses all the data but the other methods only use a half.

The results show that FedCPQR and FedGCUR consistently achieve the best or near-best performance across most cases, demonstrating the effectiveness of the selected data and features. The performance trends are largely consistent between i.i.d. and non-i.i.d. settings. Notably, FedGCUR surpasses FedCPQR in several cases, highlighting the importance of jointly selecting both data and features. This is because harmful or noisy data may exist in the dataset; using such data not only increases communication and computation overhead, but can also degrade model performance. We also observe that FedGCUR benefits from higher target ranks, likely because row selection is performed on the already-selected columns, and a larger rank provides richer information for this step. Finally, the performance of baseline methods varies across datasets, implying the need for developing more specialized data and feature selection strategies for federated learning in the future. We

Table 1: The decomposition results precision comparison between FedCPQR and SciPy.qr function.

| Dataset ID | Pivot Exactly Match | | | Max Principal Angle | | | Projection Dist. (Frobenius) | | |
|---|---|---|---|---|---|---|---|---|---|
| | k=10 | k=50 | k=100 | k=10 | k=50 | k=100 | k=10 | k=50 | k=100 |
| 1022 | True | True | True | 1.19e-15 | 5.97e-15 | 1.15e-14 | 2.27e-15 | 1.20e-14 | 2.52e-14 |
| 1041 | True | True | True | 1.02e-14 | 1.02e-14 | 1.02e-14 | 1.87e-15 | 1.35e-14 | 2.53e-14 |
| 40923 | True | True | True | 1.36e-14 | 2.23e-14 | 2.58e-14 | 6.77e-15 | 8.93e-15 | 1.13e-14 |
| 41082 | True | True | True | 1.13e-15 | 3.91e-15 | 9.50e-15 | 1.60e-15 | 9.70e-15 | 5.18e-14 |
| 41159 | True | True | True | 1.84e-15 | 2.07e-15 | 2.76e-15 | 1.37e-15 | 4.44e-15 | 8.32e-15 |
| 43985 | True | True | True | 1.53e-14 | 1.57e-14 | 1.86e-14 | 3.82e-15 | 5.96e-15 | 7.58e-15 |

Table 2: Task performance comparison of different data and feature selection methods on i.i.d. and non-i.i.d. split datasets. The mean and std. accuracy (%) are reported in the table. The best-performing method and all statistically comparable methods are highlighted in bold.

| ID | Rank | All data | FedCPQR | FedGCUR | Coreset.R | Coreset.V | Lever.R | Lever.V | Rand.R | Rand.V |
|---|---|---|---|---|---|---|---|---|---|---|
| | | | Performance on IID Split Data | | | | | | | |
| | 10 | 92.4±0.6 | **90.0±0.0** | 90.0±0.0 | 90.0±0.0 | 90.0±0.0 | 90.0±0.0 | 90.0±0.0 | 90.0±0.0 | 90.0±0.0 |
| 1022 | 50 | 92.4±0.6 | **90.0±0.0** | 90.0±0.0 | 90.0±0.0 | 90.0±0.0 | 90.0±0.0 | 90.0±0.0 | 90.0±0.0 | 90.0±0.0 |
| | 100 | 92.4±0.6 | **90.0±0.0** | **90.0±0.1** | **90.0±0.0** | **90.0±0.0** | **90.0±0.0** | **90.0±0.0** | **90.0±0.0** | **90.0±0.0** |
| | 10 | 58.8±1.3 | **11.1±1.6** | **10.4±2.0** | 10.5±0.0 | 10.5±0.0 | 10.5±0.0 | 10.6±0.1 | 10.2±0.0 | **11.1±0.0** |
| 1041 | 50 | 58.8±1.3 | **15.0±2.4** | **12.1±3.3** | 10.5±0.0 | 10.5±0.0 | 10.5±0.1 | 10.7±0.0 | 10.1±0.1 | 9.8±0.1 |
| | 100 | 58.8±1.3 | **15.4±3.4** | **17.5±2.4** | 11.5±0.6 | 11.5±0.6 | 14.8±0.8 | 10.1±0.1 | 15.3±0.6 | 12.2±0.6 |
| | 10 | 8.9±1.1 | **2.3±0.4** | **2.3±0.5** | **2.2±0.0** | **2.2±0.0** | **2.2±0.0** | **2.2±0.0** | **2.2±0.0** | **2.2±0.0** |
| 40923 | 50 | 8.9±1.1 | **2.6±0.5** | **2.2±0.5** | **2.2±0.0** | **2.2±0.0** | **2.2±0.0** | **2.2±0.0** | **2.2±0.0** | **2.2±0.0** |
| | 100 | 8.9±1.1 | **2.7±0.5** | **2.6±0.2** | 2.1±0.2 | 2.1±0.2 | **2.5±0.1** | 2.2±0.0 | 2.3±0.1 | 2.3±0.1 |
| | 10 | 60.0±0.8 | **26.4±3.8** | 20.9±7.0 | 17.2±0.7 | 17.2±0.7 | 19.1±0.5 | 16.3±0.6 | **28.5±1.4** | 24.1±1.0 |
| 41082 | 50 | 60.0±0.8 | 28.3±6.4 | **38.6±4.5** | 20.0±0.6 | 20.0±0.6 | 19.1±0.4 | 18.3±0.5 | 34.5±1.0 | 21.3±2.4 |
| | 100 | 60.0±0.8 | 42.2±4.7 | **46.9±2.8** | 25.7±1.2 | 25.7±1.2 | 37.2±1.1 | 29.9±0.6 | 37.0±1.0 | **46.2±0.8** |
| | 10 | 60.8±0.7 | **59.2±0.6** | 56.8±1.2 | **59.1±0.2** | **59.1±0.2** | 58.9±0.3 | **59.4±0.3** | 57.9±0.3 | 58.4±0.4 |
| 41159 | 50 | 60.8±0.7 | **59.0±0.6** | **58.8±0.5** | 58.2±0.5 | 58.2±0.5 | **58.7±0.2** | 58.6±0.3 | 53.6±1.3 | 56.2±0.7 |
| | 100 | 60.8±0.7 | 57.2±1.2 | **58.6±1.1** | 57.3±0.6 | 57.3±0.6 | **59.0±0.2** | 57.1±0.6 | 58.0±0.4 | 57.5±0.5 |
| | 10 | 78.7±1.2 | **15.8±2.7** | 12.7±1.6 | 14.3±0.6 | 14.3±0.6 | 12.2±0.6 | **16.9±0.8** | 14.8±1.0 | 15.4±0.8 |
| 43985 | 50 | 78.7±1.2 | **26.8±2.7** | 23.2±1.3 | 17.8±0.4 | 17.8±0.4 | 16.6±0.9 | 17.1±1.2 | 15.6±0.6 | 17.2±1.3 |
| | 100 | 78.7±1.2 | 35.7±1.8 | **37.9±2.0** | 24.4±0.8 | 24.4±0.8 | 21.2±0.5 | 21.5±0.9 | 21.4±1.0 | 24.8±2.0 |
| | | | Performance on Non-IID Split Data | | | | | | | |
| ID | Rank | All data | FedCPQR | FedGCUR | Coreset.R | Coreset.V | Lever.R | Lever.V | Rand.R | Rand.V |
| | 10 | 91.7±0.2 | **90.0±0.0** | 90.0±0.0 | 90.0±0.0 | 90.0±0.0 | 90.0±0.0 | 90.0±0.0 | 90.0±0.0 | 90.0±0.0 |
| 1022 | 50 | 91.7±0.2 | **90.0±0.0** | 90.0±0.0 | 90.0±0.0 | 90.0±0.0 | 90.0±0.0 | 90.0±0.0 | 90.0±0.0 | 90.0±0.0 |
| | 100 | 91.7±0.2 | **90.0±0.0** | 90.0±0.0 | 90.0±0.0 | 90.0±0.0 | 90.0±0.0 | 90.0±0.0 | 90.0±0.0 | 90.0±0.0 |
| | 10 | 58.8±1.4 | **11.0±1.6** | **10.3±2.1** | 10.5±0.0 | 10.5±0.0 | 10.5±0.0 | 10.6±0.1 | 10.2±0.0 | **10.9±0.3** |
| 1041 | 50 | 58.8±1.4 | **15.8±2.1** | **12.8±2.7** | 10.5±0.0 | 10.5±0.0 | 10.5±0.1 | 10.7±0.0 | 10.1±0.1 | 9.8±0.1 |
| | 100 | 58.8±1.4 | **16.3±2.2** | **17.6±1.4** | 11.0±0.4 | 11.0±0.4 | 14.5±0.9 | 10.1±0.1 | 15.1±0.4 | 12.0±0.4 |
| | 10 | 9.3±0.2 | **2.1±0.3** | **2.3±0.4** | **2.2±0.0** | **2.2±0.0** | **2.2±0.0** | **2.2±0.0** | **2.2±0.0** | **2.2±0.0** |
| 40923 | 50 | 9.3±0.2 | **2.5±0.5** | **2.3±0.5** | **2.2±0.0** | **2.2±0.0** | **2.2±0.0** | **2.2±0.0** | **2.2±0.0** | **2.2±0.0** |
| | 100 | 9.3±0.2 | 2.5±0.3 | **2.8±0.3** | 2.3±0.2 | 2.3±0.2 | 2.6±0.1 | 2.2±0.1 | 2.3±0.1 | 2.3±0.1 |
| | 10 | 59.8±0.7 | 26.8±4.1 | **31.9±6.3** | 17.7±1.0 | 17.7±1.0 | 20.0±0.5 | 17.6±1.1 | **28.7±0.7** | 24.5±1.0 |
| 41082 | 50 | 59.8±0.7 | 29.7±5.7 | **39.3±1.5** | 22.1±2.6 | 22.1±2.6 | 18.6±0.2 | 17.9±0.7 | 34.6±0.8 | 20.5±1.0 |
| | 100 | 59.8±0.7 | 41.8±4.8 | 41.5±3.0 | 26.1±1.7 | 26.1±1.7 | 35.7±0.8 | 31.2±1.2 | 36.9±0.5 | **46.1±0.6** |
| | 10 | 60.7±0.6 | 59.1±0.6 | **60.0±0.1** | 59.1±0.3 | 59.1±0.3 | 58.6±0.2 | 59.1±0.3 | 57.8±0.3 | 58.5±0.5 |
| 41159 | 50 | 60.7±0.6 | 59.0±0.5 | **60.0±0.0** | 58.1±0.5 | 58.1±0.5 | 58.3±0.5 | 58.4±0.3 | 53.6±1.3 | 55.9±0.9 |
| | 100 | 60.7±0.6 | 57.2±1.2 | **60.0±0.0** | 57.1±0.6 | 57.1±0.6 | 58.8±0.2 | 56.8±0.6 | 57.8±0.5 | 57.5±0.6 |
| | 10 | 73.3±2.5 | **14.6±1.9** | 11.7±2.7 | 12.8±1.4 | 12.8±1.4 | 11.1±1.6 | **14.9±1.7** | **14.1±0.7** | 14.2±1.0 |
| 43985 | 50 | 73.3±2.5 | **25.3±2.7** | 16.9±2.1 | **17.7±2.7** | **17.7±2.7** | 16.8±1.3 | 16.3±2.3 | 16.9±1.8 | 16.5±2.1 |
| | 100 | 73.3±2.5 | **32.7±2.9** | 25.9±0.8 | 24.4±1.2 | 24.4±1.2 | 21.1±0.8 | 21.0±1.1 | 20.2±1.5 | 21.7±1.4 |

note that the performance of all methods on dataset 40923 is relatively low. This is primarily due to the use of a limited number of communication rounds (10), which we fix across all datasets to ensure fairness in comparison. Since our focus in this experiment is on relative improvements, we retain the same training configuration throughout. For completeness, we also report per-client and per-class performance metrics under non-i.i.d. splits in the appendix, which show that FedGCUR does not introduce adverse fairness effects compared with the strongest baselines.

**Efficiency of FedCPQR and FedGCUR**  We report the log-scale running times (in seconds) of FedQR, FedCPQR, and FedGCUR under varying numbers of parties and target ranks in Figure 1. In the experiment varying the number of parties, the target rank is fixed at 50; in the experiment varying the target rank, the number of parties is fixed at 10. Each result is averaged over five runs, with error bars showing standard deviations. Since FedQR is independent of the target rank, it is excluded from the varying-rank experiment. The results show that the running times of FedCPQR and FedGCUR remain relatively low and do not increase substantially as either the number of parties or the target rank grows. Furthermore, both methods generally run faster than the FedQR baseline,

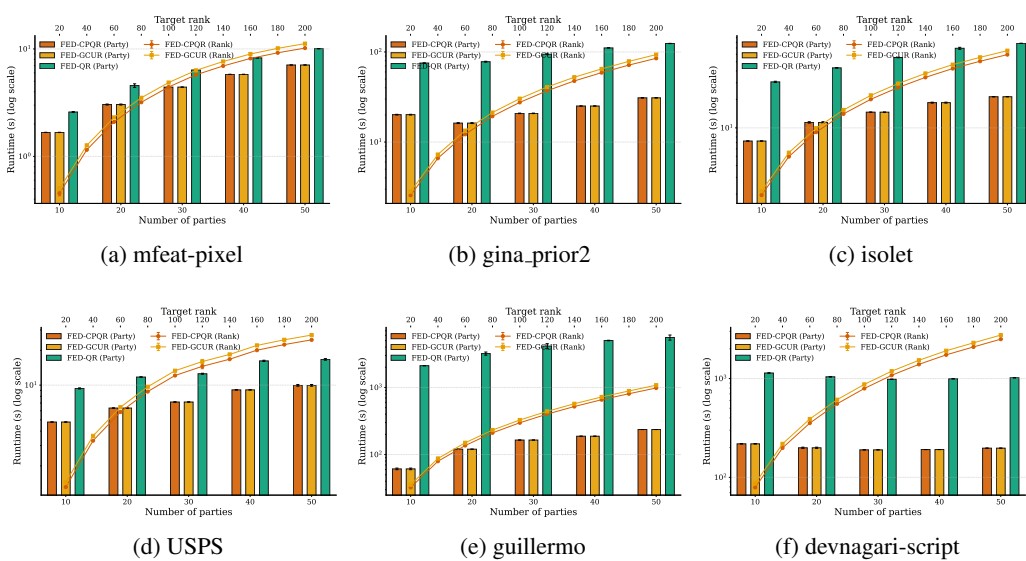

(a) mfeat-pixel     (b) gina_prior2     (c) isolet

(d) USPS     (e) guillermo     (f) devnagari-script

Figure 1: Log-scale running times (in seconds) of FedQR, FedCPQR, and FedGCUR with varying numbers of parties and ranks. Error bars indicating standard deviations of 5 runs.

highlighting their efficiency in handling high-dimensional data with relatively small target ranks in FL. In the varying-rank experiment, the running times of FedCPQR and FedGCUR are comparable, with FedGCUR being slightly more expensive. This suggests that the computational overhead of FedGCUR remains reasonable, making it practical for both data and feature selection. These empirical measurements are consistent with the analytical communication-cost formulas derived in Sec. 3.2.1: the number of aggregated scalars per iteration scales as $\mathcal{O}(d^2)$ and is independent of the sample size and number of clients (assuming an efficient implementation of secure aggregation), in contrast to iterative training methods such as FedAvg that must repeatedly transmit high-dimensional gradients over many rounds.

## 5 CONCLUSION

In this paper, we proposed a practical framework for joint data and feature selection in federated learning based on generalized CUR decomposition. Our approach combines FedCPQR that exactly reproduces centralized CPQR pivoting in a horizontal FL setting via secure aggregation of norms and inner products, with FedGCUR, which uses the resulting global pivot order to define a shared feature set and per-silo row selections under local budget constraints. We analyzed the communication, computation, and correctness properties of FedCPQR, showed that it matches the centralized modified Gram–Schmidt CPQR under exact aggregation, and established an upper bound on the reconstruction error of FedGCUR that captures the interaction between global column and block-diagonal row projectors. Extensive experiments across diverse datasets and both i.i.d. and non-i.i.d. splits demonstrated that FedGCUR achieves competitive or superior performance relative to strong data and feature selection baselines, while operating under realistic communication budgets where training on all data is infeasible. Our work highlights the importance of treating data and feature selection as a coupled problem in federated learning and shows that generalized CUR provides a useful lens for doing so under cross-silo constraints. Several directions remain open for future research, including adaptive rank and budget selection, extensions to more complex models and tasks, and systematic integration with robustness mechanisms against noisy or adversarial clients and with partial-participation protocols.

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

## A  LLM Usage Statement

The authors declare that Large Language Models (LLMs) were utilized in the preparation of this manuscript. Their use was limited to linguistic refinement, including polishing wording, improving sentence structure for clarity, and correcting spelling. Additionally, LLMs were employed to identify and correct minor errors in mathematical symbols and formulas. No part of the conceptualization, methodological design, data analysis, or substantive interpretation was generated by LLMs. The au-thors retain full responsibility for the content, accuracy, and originality of all scientific contributions presented in this paper.

## B  Reproducibility Statement

All experiments in this work were conducted using fixed random seeds to ensure reproducibility of the reported results. The experimental settings, including model configurations, training procedures, and evaluation protocols, were kept consistent across runs. To further support reproducibility, we will make the source code and implementation details publicly available at a later stage.

## C  Proof of Theorem 1

We restate Theorem 1 first, and then present the proof.

**Theorem 1.** *Assume exact secure aggregation of all required inner products and squared norms. The pivot choices and updates of FedCPQR are identical to those of centralized modified Gram–Schmidt based CPQR on the global matrix.*

*Proof.* All quantities used by modified Gram–Schmidt based CPQR can be expressed as sums over local contributions:

$$\|U(:,j)\|_2^2 = \sum_c \|U_c(:,j)\|_2^2, \qquad Q(:,i)^\top U(:,j) = \sum_c Q_c(:,i)^\top U_c(:,j).$$

Secure aggregation returns these exact global scalars. Hence, FedCPQR selects the same pivot at each iteration, sets the same $R_{ij}$, and updates the same residuals. The resulting $P$, $Q$, $R$, and derived projectors coincide with the centralized modified Gram–Schmidt based CPQR. $\qquad\square$

## D    PROOF OF THEOREM 2

We first introduce the following lemma.

**Lemma 1** (Lemma 3.2 (Dong & Martinsson, 2021)). *Let $X \in \mathbb{R}^{\ell \times n}$ and run CPQR on $X$ to select $k$ pivot columns. Write the CPQR as*

$$X\Pi = Q\,[\,T_1 \ T_2\,], \tag{9}$$

*where $\Pi = [\Pi_c,\, \Pi_{cc}]$ is a permutation with $\Pi_c \in \mathbb{R}^{n \times k}$ extracting the first $k$ pivots, $Q \in \mathbb{R}^{\ell \times k}$ has orthonormal columns, and $T_1 \in \mathbb{R}^{k \times k}$ is nonsingular upper triangular. Define the right-acting oblique projector in the column-index space of $X$ by*

$$P_X = \Pi_c\,(X\Pi_c)^\dagger\,X = \Pi_c\,T_1^{-1}Q^\top X \in \mathbb{R}^{n \times n}. \tag{10}$$

*Then*

$$\|I_n - P_X\| \le \sqrt{1 + (n - k)\,4^{\,k-1}}\,. \tag{11}$$

We restate theorem 2 here again for easier reading and present the proof.

**Theorem 2.** *Let $A \in \mathbb{R}^{n \times d}$ be partitioned by rows into $s$ parties, $A = \big[A_1^\top, \ldots, A_s^\top\big]^\top$, and $A_c \in \mathbb{R}^{n_c \times d}$. Let $\widehat{A}_{\mathrm{loc}}$ be the FedGCUR reconstruction defined above, $k$ and $r_c$ be the number of selected columns and rows in party $c$, respectively. Then*

$$\|A - \widehat{A}_{\mathrm{loc}}\| \le \sqrt{1 + (d - k)\,4^{\,k-1}}\ \sigma_{k+1}(A) + \max_{1 \le c \le s} \sqrt{1 + (n_c - r_c)\,4^{\,r_c-1}}\ \sigma_{r_c+1}(C_c),$$

$$\|A - \widehat{A}_{\mathrm{loc}}\|_F \le \sqrt{1 + (d - k)\,4^{\,k-1}}\Big(\sum_{j>k} \sigma_j^2(A)\Big)^{1/2} + \Big(\sum_{c=1}^{s}\big[1 + (n_c - r_c)\,4^{\,r_c-1}\big]\sum_{j>r_c} \sigma_j^2(C_c)\Big)^{1/2}.$$

*Proof.* Since $P_C$ and $P_R$ are idempotent and $P_C$ is an orthogonal projector, we obtain

$$\|A - \widehat{A}_{\mathrm{loc}}\|_2 \le \|(I - P_C)A\|_2 + \|P_C A(I - P_R)\|_2 \le \|(I - P_C)A\|_2 + \|A(I - P_R)\|_2, \tag{12}$$

$$\|A - \widehat{A}_{\mathrm{loc}}\|_F^2 = \|(I - P_C)A\|_F^2 + \|P_C A(I - P_R)\|_F^2 \le \|(I - P_C)A\|_F^2 + \|A(I - P_R)\|_F^2. \tag{13}$$

The Frobenius equality above uses that the column spaces $\mathrm{range}(I - P_C)$ and $\mathrm{range}(P_C)$ are orthogonal.

Let $X \in \mathbb{R}^{n \times d}$ be the matrix on which CPQR is run to pick the $k$ column pivots. Write the CPQR as $X\Pi = Q\,[\,R_1 \ R_2\,]$ where $\Pi = [\Pi_c, \Pi_{cc}]$ and $R_1 \in \mathbb{R}^{k \times k}$ is nonsingular. Define the right–acting oblique projector in the column–index space

$$P_X := \Pi_c\,(X\Pi_c)^\dagger X = \Pi_c R_1^{-1} Q^\top X \in \mathbb{R}^{d \times d}. \tag{14}$$

Then $X(I - P_X) = 0$ and

$$\|(I - P_C)A\|_\xi \le \|I - P_X\|\,\|A(I - P_k)\|_\xi, \qquad \xi \in \{2, F\}, \tag{15}$$

where $P_k$ is the orthogonal projector onto the $k$ selected coordinates. By Lemma 1, we have $\|I - P_X\| \le \sqrt{1 + (d - k)\,4^{\,k-1}}$. By Eckart–Young–Mirsky theorem (Eckart & Young, 1936), we have $\|A(I - P_k)\|_2 = \sigma_{k+1}(A)$ and $\|A(I - P_k)\|_F = (\sum_{j>k} \sigma_j^2(A))^{1/2}$. This yields

$$\|(I - P_C)A\|_2 \le \sqrt{1 + (d - k)\,4^{\,k-1}}\ \sigma_{k+1}(A), \tag{16}$$

$$\|(I - P_C)A\|_F \le \sqrt{1 + (d - k)\,4^{\,k-1}}\ \Big(\sum_{j>k} \sigma_j^2(A)\Big)^{1/2}. \tag{17}$$

Fix a client $c$. Apply CPQR to $C_c^\top \in \mathbb{R}^{d \times n_c}$ to select $r_c$ rows, and let $P_{L,c} \in \mathbb{R}^{n_c \times n_c}$ be the associated left–acting oblique projector in the row–index space (constructed as in Lemma 1, with $(n, k) \leftarrow (n_c, r_c)$). Then

$$\|I_{n_c} - P_{L,c}\| \leq \sqrt{1 + (n_c - r_c)\, 4^{\,r_c-1}}. \tag{18}$$

Let $P_{r_c}^{(c)}$ be the orthogonal projector onto the top $r_c$ right singular vectors of $C_c$. Again, by Eckart–Young–Mirsky theorem, we have

$$\|C_c(I - P_{r_c}^{(c)})\|_2 = \sigma_{r_c+1}(C_c), \qquad \|C_c(I - P_{r_c}^{(c)})\|_F = \Big( \sum_{j > r_c} \sigma_j^2(C_c) \Big)^{1/2}. \tag{19}$$

By a similar argument in the column analysis with transposed roles, we have

$$\|(I - P_{L,c})\, C_c\|_\xi \leq \|I - P_{L,c}\|\, \|C_c(I - P_{r_c}^{(c)})\|_\xi, \qquad \xi \in \{2, F\}. \tag{20}$$

Since $P_{R,c}$ is the orthogonal projector onto the selected row space, $C_c(I - P_{R,c}) = 0$, and hence for any $X_c$, $A_c(I - P_{R,c}) = (A_c - C_c)(I - P_{R,c})$, so that $\|A_c(I - P_{R,c})\|_\xi \leq \|(I - P_{L,c})\, C_c\|_\xi$. Combining the displays and using Eq. (18),

$$\|A_c(I - P_{R,c})\|_\xi \leq \sqrt{1 + (n_c - r_c)\, 4^{\,r_c-1}} \times \begin{cases} \sigma_{r_c+1}(C_c), & \xi = 2, \\ \big( \sum_{j > r_c} \sigma_j^2(C_c) \big)^{1/2}, & \xi = F. \end{cases} \tag{21}$$

Stacking block rows yields

$$A(I - P_R) = \begin{bmatrix} A_1(I - P_{R,1}) \\ \vdots \\ A_s(I - P_{R,s}) \end{bmatrix}.$$

Hence

$$\|A(I - P_R)\|_2^2 \leq \sum_{c=1}^{s} \|A_c(I - P_{R,c})\|_2^2, \tag{22}$$

$$\|A(I - P_R)\|_F^2 = \sum_{c=1}^{s} \|A_c(I - P_{R,c})\|_F^2. \tag{23}$$

Insert Eq. (21) into Eq. (22)–Eq. (23) to obtain

$$\|A(I - P_R)\|_2 \leq \left( \sum_{c=1}^{s} \big[ 1 + (n_c - r_c)\, 4^{\,r_c-1} \big]\, \sigma_{r_c+1}^2(C_c) \right)^{1/2}, \tag{24}$$

$$\|A(I - P_R)\|_F^2 \leq \sum_{c=1}^{s} \big[ 1 + (n_c - r_c)\, 4^{\,r_c-1} \big] \sum_{j > r_c} \sigma_j^2(C_c). \tag{25}$$

Combine Eq. (12) with Eq. (16) and Eq. (24) to get Eq. (7); combine Eq. (13) with Eq. (17) and Eq. (25) to get Eq. (8). $\qquad\square$

# E    RESULTS OF DIFFERENT NUMBERS OF CLIENTS

We report the performance comparison of 50 clients in Table 5. The experimental setting strictly adheres to the configuration detailed in Table 4 of the original paper.

It is observed that the comparison results are analogous to the scenario involving 10 clients. The absolute performance metrics are slightly worse than the results reported in Table 4, a phenomenon we attribute to the increased learning difficulty introduced by the greater number of participating clients. Despite this minor variance in absolute values, the relative performance rankings remain fundamentally consistent with the 10-client setting. Specifically, our proposed method consistently achieves superior performance, often securing the best ranking across the evaluated metrics. This empirical evidence demonstrates the robust and consistent effectiveness of our approach, even when applied to a more challenging setting involving a significantly larger population of clients.

Table 3: Key differences between FedQR and FedCPQR.

|  | **FedQR** (Hartebrodt & Röttger, 2023) | **FedCPQR** (Proposed in this paper) |
|---|---|---|
| Rows of $A$ | Kept local | Kept local |
| Factor $Q_c$ | Kept local | Kept local |
| Shared factor | $R$ | $R, P$ |
| Also revealed | Global norms / inner products | Same plus pivot residual norms |
| Implied leakage | $A^\top A = R^\top R$ | $A^\top A = PR^\top RP^\top$ |
| Special-case risk | Raw data may leak if only 2 parties | Same |

Table 4: Datasets Summary.

| Dataset | OpenML ID | # Train data | # Test data | # Features | # Classes |
|---|---|---|---|---|---|
| mfeat-pixel | 1022 | 1600 | 400 | 240 | 2 |
| gina_prior2 | 1041 | 2774 | 694 | 784 | 10 |
| devnagari-script | 40923 | 73600 | 18400 | 1024 | 46 |
| USPS | 41082 | 7438 | 1860 | 256 | 10 |
| guillermo | 41159 | 16000 | 4000 | 4296 | 2 |
| isolet | 43985 | 6237 | 1560 | 613 | 26 |

## F  PER-CLIENT PERFORMANCE

To complement the aggregate accuracy results in Section 4, we report here per-client and per-class performance metrics under Dirichlet non-i.i.d. splits. Here, we report the results of the random seed 0. For each dataset and selection method, we compute macro-averaged accuracy across clients, the standard deviation of per-client accuracies, and per-class recall aggregated over silos. These detailed tables provide a finer-grained view of fairness and heterogeneity effects.

Overall, the patterns corroborate the main findings: FedGCUR matches or improves the macro-average accuracy of strong baselines while not increasing the spread of per-client performance. In particular, minority or low-data clients under non-i.i.d. splits do not suffer disproportionate degradation compared with other parties when FedGCUR is applied.

## G  FEDERATED LEARNING SPECIFICATIONS

- **Model architecture**: MLP with two hidden layers of 64 and 32 units.
- **Optimizer settings**: Local training uses a learning rate of $0.01$ and $5$ local epochs per communication round.
- **Communication schedule**: Global aggregation runs for $50$ rounds by default.
- **Output configuration**: Classification heads predict $|\mathcal{Y}|$ classes after label encoding.
- **Participation**: All available clients contribute their gradients at each round.

Table 5: Task performance comparison of 50 clients of different data and feature selection methods on i.i.d. and non-i.i.d. split datasets. The mean and std. accuracy (%) are reported in the table. The best-performing method and all statistically comparable methods are highlighted in bold.

| ID | Rank | All data | FedCPQR | FedGCUR | Coreset.R | Coreset.V | Lever.R | Lever.V | Rand.R | Rand.V |
|---|---|---|---|---|---|---|---|---|---|---|
| | | | | Performance on IID Split Data | | | | | | |
| | 10 | 90.4±1.3 | **90.0±0.0** | 90.0±0.0 | 90.0±0.0 | 90.0±0.0 | 90.0±0.0 | 90.0±0.0 | 90.0±0.0 | 90.0±0.0 |
| 1022 | 50 | 90.4±1.3 | **90.0±0.0** | 90.0±0.0 | 90.0±0.0 | 90.0±0.0 | 90.0±0.0 | 90.0±0.0 | 90.0±0.0 | 90.0±0.0 |
| | 100 | 90.4±1.3 | **90.1±0.2** | **90.3±0.4** | **90.0±0.1** | **90.0±0.1** | 90.0±0.0 | 90.0±0.0 | 90.0±0.0 | 90.0±0.0 |
| | 10 | 56.7±1.4 | **12.3±3.1** | **13.0±2.5** | 9.1±0.0 | 9.1±0.0 | 9.8±0.0 | 9.8±0.8 | 10.5±0.0 | 10.5±0.0 |
| 1041 | 50 | 56.7±1.4 | 12.6±3.1 | **15.7±1.4** | 9.9±0.2 | 9.9±0.2 | 9.9±0.2 | 10.5±0.0 | 9.1±0.0 | 9.2±0.1 |
| | 100 | 56.7±1.4 | 15.2±2.3 | **18.7±3.2** | 15.4±0.5 | 15.4±0.5 | 14.2±1.0 | 13.2±0.9 | **16.7±0.7** | 11.4±0.5 |
| | 10 | 10.0±0.3 | 2.2±0.2 | **2.6±0.4** | 2.2±0.0 | 2.2±0.0 | 2.2±0.0 | 2.2±0.0 | 2.2±0.0 | 2.2±0.0 |
| 40923 | 50 | 10.0±0.3 | **2.9±0.4** | 2.3±0.3 | **2.2±0.0** | **2.2±0.0** | **2.2±0.0** | **2.2±0.0** | **2.2±0.0** | **2.2±0.0** |
| | 100 | 10.0±0.3 | **2.9±0.6** | **3.3±0.6** | 2.4±0.1 | 2.4±0.1 | 2.6±0.1 | 2.1±0.1 | 2.2±0.1 | 2.2±0.1 |
| | 10 | 56.6±1.8 | **26.1±5.8** | 13.6±2.8 | 16.4±0.7 | 16.4±0.7 | 14.3±0.3 | 10.7±0.3 | 13.8±0.6 | **23.9±0.9** |
| 41082 | 50 | 56.6±1.8 | **36.4±4.1** | 28.1±3.3 | 26.1±0.6 | 26.1±0.6 | 18.2±0.7 | 13.1±0.4 | **26.7±1.0** | 25.8±0.4 |
| | 100 | 56.6±1.8 | 38.7±1.9 | **41.5±2.9** | 26.3±1.0 | 26.3±1.0 | 26.3±0.7 | 27.1±0.5 | 40.0±1.3 | **43.1±1.4** |
| | 10 | 60.1±0.4 | 58.2±0.8 | **59.8±0.2** | 58.5±0.2 | 58.5±0.2 | **59.9±0.0** | 57.2±0.3 | 57.1±0.5 | 58.8±0.3 |
| 41159 | 50 | 60.1±0.4 | 56.1±0.6 | 57.4±0.9 | 56.0±0.2 | 56.0±0.2 | 58.6±0.2 | **59.5±0.1** | 58.3±0.3 | 57.6±0.5 |
| | 100 | 60.1±0.4 | 56.4±0.9 | 56.9±1.0 | 56.5±0.6 | 56.5±0.6 | **59.8±0.1** | 57.5±0.6 | 59.2±0.5 | 58.5±0.5 |
| | 10 | 34.1±2.2 | 4.3±1.1 | 5.7±2.2 | 3.9±0.2 | 3.9±0.2 | **8.1±0.8** | 7.5±0.6 | 6.0±0.2 | **8.8±0.5** |
| 43985 | 50 | 34.1±2.2 | 6.2±1.5 | 7.3±1.8 | 7.1±0.3 | 7.1±0.3 | **9.5±0.6** | **9.8±0.7** | **10.0±0.7** | 9.2±0.4 |
| | 100 | 34.1±2.2 | 6.2±1.5 | 8.3±0.9 | 10.3±0.7 | 10.3±0.7 | 10.1±0.6 | 10.5±0.3 | **12.5±0.9** | 8.3±0.7 |
| | | | | Performance on Non-IID Split Data | | | | | | |
| | 10 | 90.0±0.0 | **90.0±0.0** | 90.0±0.0 | 90.0±0.0 | 90.0±0.0 | 90.0±0.0 | 90.0±0.0 | 90.0±0.0 | 90.0±0.0 |
| 1022 | 50 | 90.0±0.0 | **90.0±0.0** | 90.0±0.0 | 90.0±0.0 | 90.0±0.0 | 90.0±0.0 | 90.0±0.0 | 90.0±0.0 | 90.0±0.0 |
| | 100 | 90.0±0.0 | **90.0±0.0** | **90.1±0.2** | 89.9±0.2 | 89.9±0.2 | **90.0±0.0** | **90.0±0.0** | **90.0±0.0** | **90.0±0.0** |
| | 10 | 56.6±1.2 | **11.5±2.0** | **12.1±2.8** | 9.9±0.7 | 9.9±0.7 | 9.8±0.0 | **10.7±0.0** | **10.5±0.0** | **10.5±0.0** |
| 1041 | 50 | 56.6±1.2 | **12.9±3.4** | **15.6±1.7** | 9.9±0.2 | 9.9±0.2 | 9.9±0.2 | 10.5±0.0 | 9.1±0.0 | 9.2±0.1 |
| | 100 | 56.6±1.2 | 14.4±2.0 | **18.0±2.6** | 14.6±0.7 | 14.6±0.7 | 13.6±0.9 | 12.9±0.8 | **16.7±0.8** | 11.0±0.1 |
| | 10 | 10.1±0.1 | 2.2±0.1 | **2.7±0.3** | 2.2±0.0 | 2.2±0.0 | 2.2±0.0 | 2.2±0.0 | 2.2±0.0 | 2.2±0.0 |
| 40923 | 50 | 10.1±0.1 | **3.0±0.5** | 2.5±0.4 | 2.2±0.0 | 2.2±0.0 | 2.2±0.0 | 2.2±0.0 | 2.2±0.0 | 2.2±0.0 |
| | 100 | 10.1±0.1 | 2.8±0.6 | **3.6±0.4** | 2.4±0.1 | 2.4±0.1 | 2.6±0.1 | 2.1±0.1 | 2.2±0.1 | 2.2±0.1 |
| | 10 | 56.1±0.7 | **23.7±3.5** | 14.5±4.0 | 16.0±0.7 | 16.0±0.7 | 14.0±0.1 | 10.8±0.3 | 13.7±0.6 | **23.5±0.9** |
| 41082 | 50 | 56.1±0.7 | **36.1±3.7** | 28.3±3.1 | 25.9±0.9 | 25.9±0.9 | 18.4±0.7 | 13.8±0.8 | **26.3±0.9** | 25.7±0.6 |
| | 100 | 56.1±0.7 | 38.7±2.0 | **43.0±6.9** | 26.7±3.1 | 26.7±3.1 | 25.5±0.8 | 27.8±0.8 | 40.4±0.8 | **43.3±1.3** |
| | 10 | 59.8±0.2 | 58.2±0.8 | **60.0±0.0** | 58.6±0.2 | 58.6±0.2 | **59.9±0.1** | 56.5±0.4 | 57.2±0.7 | 59.0±0.3 |
| 41159 | 50 | 59.8±0.2 | 56.1±0.6 | **59.6±0.2** | 56.3±0.4 | 56.3±0.4 | 58.1±0.2 | 59.2±0.1 | 58.3±0.3 | 57.7±0.4 |
| | 100 | 59.8±0.2 | 56.4±0.9 | **58.9±1.0** | 56.6±0.7 | 56.6±0.7 | **59.6±0.2** | 56.7±0.6 | **59.2±0.5** | 58.7±0.6 |
| | 10 | 34.9±0.8 | 4.6±1.3 | 5.6±1.9 | 4.1±0.2 | 4.1±0.2 | 7.9±0.6 | 7.4±0.7 | 5.8±0.5 | **8.7±0.6** |
| 43985 | 50 | 34.9±0.8 | 6.0±1.5 | 6.7±1.7 | 7.0±0.3 | 7.0±0.3 | 9.2±0.5 | **9.6±0.8** | **10.1±0.7** | 8.9±0.4 |
| | 100 | 34.9±0.8 | 6.3±1.2 | 8.0±1.0 | 10.1±0.8 | 10.1±0.8 | 10.0±0.6 | 10.6±0.4 | **12.4±0.8** | 8.2±0.8 |

Table 6: Per-client Downstream Performance for 1022

| Partition | rank | client | FedCPQR | FedGCUR | K.-R. | K.-V. | L.-R. | L.-V. | R.-R. | R.-V. |
|---|---|---|---|---|---|---|---|---|---|---|
| IID | 10 | 0 | 90.0% | 90.0% | 90.0% | 90.0% | 90.0% | 90.0% | 90.0% | 90.0% |
| | | 1 | 90.0% | 90.0% | 90.0% | 90.0% | 90.0% | 90.0% | 90.0% | 90.0% |
| | | 2 | 90.0% | 90.0% | 90.0% | 90.0% | 90.0% | 90.0% | 90.0% | 90.0% |
| | | 3 | 90.0% | 90.0% | 90.0% | 90.0% | 90.0% | 90.0% | 90.0% | 90.0% |
| | | 4 | 90.0% | 90.0% | 90.0% | 90.0% | 90.0% | 90.0% | 90.0% | 90.0% |
| | | 5 | 90.0% | 90.0% | 90.0% | 90.0% | 90.0% | 90.0% | 90.0% | 90.0% |
| | | 6 | 90.0% | 90.0% | 90.0% | 90.0% | 90.0% | 90.0% | 90.0% | 90.0% |
| | | 7 | 90.0% | 90.0% | 90.0% | 90.0% | 90.0% | 90.0% | 90.0% | 90.0% |
| | | 8 | 90.0% | 90.0% | 90.0% | 90.0% | 90.0% | 90.0% | 90.0% | 90.0% |
| | | 9 | 90.0% | 90.0% | 90.0% | 90.0% | 90.0% | 90.0% | 90.0% | 90.0% |
| | 50 | 0 | 90.0% | 90.0% | 90.0% | 90.0% | 90.0% | 90.0% | 90.0% | 90.0% |
| | | 1 | 90.0% | 90.0% | 90.0% | 90.0% | 90.0% | 90.0% | 90.0% | 90.0% |
| | | 2 | 90.0% | 90.0% | 90.0% | 90.0% | 90.0% | 90.0% | 90.0% | 90.0% |
| | | 3 | 90.0% | 90.0% | 90.0% | 90.0% | 90.0% | 90.0% | 90.0% | 90.0% |
| | | 4 | 90.0% | 90.0% | 90.0% | 90.0% | 90.0% | 90.0% | 90.0% | 90.0% |
| | | 5 | 90.0% | 90.0% | 90.0% | 90.0% | 90.0% | 90.0% | 90.0% | 90.0% |
| | | 6 | 90.0% | 90.0% | 90.0% | 90.0% | 90.0% | 90.0% | 90.0% | 90.0% |
| | | 7 | 90.0% | 90.0% | 90.0% | 90.0% | 90.0% | 90.0% | 90.0% | 90.0% |
| | | 8 | 90.0% | 90.0% | 90.0% | 90.0% | 90.0% | 90.0% | 90.0% | 90.0% |
| | | 9 | 90.0% | 90.0% | 90.0% | 90.0% | 90.0% | 90.0% | 90.0% | 90.0% |
| | 100 | 0 | 90.0% | 90.0% | 90.0% | 90.0% | 90.0% | 90.0% | 90.0% | 90.0% |
| | | 1 | 90.0% | 90.0% | 90.0% | 90.0% | 90.0% | 90.0% | 90.0% | 90.0% |
| | | 2 | 90.0% | 90.0% | 90.0% | 90.0% | 90.0% | 90.0% | 90.0% | 90.0% |
| | | 3 | 90.0% | 90.0% | 90.0% | 90.0% | 90.0% | 90.0% | 90.0% | 90.0% |
| | | 4 | 90.0% | 90.0% | 90.0% | 90.0% | 90.0% | 90.0% | 90.0% | 90.0% |
| | | 5 | 90.0% | 90.0% | 90.0% | 90.0% | 90.0% | 90.0% | 90.0% | 90.0% |
| | | 6 | 90.0% | 90.0% | 90.0% | 90.0% | 90.0% | 90.0% | 90.0% | 90.0% |
| | | 7 | 90.0% | 90.0% | 90.0% | 90.0% | 90.0% | 90.0% | 90.0% | 90.0% |
| | | 8 | 90.0% | 90.0% | 90.0% | 90.0% | 90.0% | 90.0% | 90.0% | 90.0% |
| | | 9 | 90.0% | 90.0% | 90.0% | 90.0% | 90.0% | 90.0% | 90.0% | 90.0% |
| Non-IID | 10 | 0 | 90.0% | 90.0% | 90.0% | 90.0% | 90.0% | 90.0% | 90.0% | 90.0% |
| | | 1 | 90.0% | 90.0% | 90.0% | 90.0% | 90.0% | 90.0% | 90.0% | 90.0% |
| | | 2 | 90.0% | 90.0% | 90.0% | 90.0% | 90.0% | 90.0% | 90.0% | 90.0% |
| | | 3 | 90.0% | 90.0% | 90.0% | 90.0% | 90.0% | 90.0% | 90.0% | 90.0% |
| | | 4 | 90.0% | 90.0% | 90.0% | 90.0% | 90.0% | 90.0% | 90.0% | 90.0% |
| | | 5 | 90.0% | 90.0% | 90.0% | 90.0% | 90.0% | 90.0% | 90.0% | 90.0% |
| | | 6 | 90.0% | 90.0% | 90.0% | 90.0% | 90.0% | 90.0% | 90.0% | 90.0% |
| | | 7 | 90.0% | 90.0% | 90.0% | 90.0% | 90.0% | 90.0% | 90.0% | 90.0% |
| | | 8 | 90.0% | 90.0% | 90.0% | 90.0% | 90.0% | 90.0% | 90.0% | 90.0% |
| | | 9 | 90.0% | 90.0% | 90.0% | 90.0% | 90.0% | 90.0% | 90.0% | 90.0% |
| | 50 | 0 | 90.0% | 90.0% | 90.0% | 90.0% | 90.0% | 90.0% | 90.0% | 90.0% |
| | | 1 | 90.0% | 90.0% | 90.0% | 90.0% | 90.0% | 90.0% | 90.0% | 90.0% |
| | | 2 | 90.0% | 90.0% | 90.0% | 90.0% | 90.0% | 90.0% | 90.0% | 90.0% |
| | | 3 | 90.0% | 90.0% | 90.0% | 90.0% | 89.7% | 90.0% | 90.0% | 90.0% |
| | | 4 | 90.0% | 90.0% | 90.0% | 90.0% | 90.0% | 90.0% | 90.0% | 90.0% |
| | | 5 | 90.0% | 90.0% | 90.0% | 90.0% | 89.7% | 90.0% | 90.0% | 90.0% |
| | | 6 | 90.0% | 90.0% | 90.0% | 90.0% | 90.0% | 90.0% | 90.0% | 90.0% |
| | | 7 | 90.0% | 90.0% | 90.0% | 90.0% | 89.7% | 90.0% | 90.0% | 90.0% |
| | | 8 | 90.0% | 90.0% | 90.0% | 90.0% | 90.0% | 90.0% | 90.0% | 90.0% |
| | | 9 | 90.0% | 90.0% | 90.0% | 90.0% | 90.0% | 90.0% | 90.0% | 90.0% |
| | 100 | 0 | 90.0% | 90.0% | 90.0% | 90.0% | 90.0% | 90.0% | 90.0% | 90.0% |
| | | 1 | 90.0% | 90.0% | 90.0% | 90.0% | 90.0% | 90.0% | 90.0% | 90.0% |
| | | 2 | 90.0% | 90.0% | 90.0% | 90.0% | 90.0% | 90.0% | 90.0% | 90.0% |
| | | 3 | 90.0% | 90.0% | 90.0% | 90.0% | 90.0% | 89.7% | 90.0% | 89.7% |
| | | 4 | 90.0% | 90.0% | 90.0% | 90.0% | 90.0% | 89.5% | 90.0% | 89.0% |
| | | 5 | 90.0% | 90.5% | 90.0% | 90.0% | 90.2% | 87.3% | 89.7% | 90.5% |
| | | 6 | 90.0% | 90.0% | 90.0% | 90.0% | 90.0% | 90.0% | 90.0% | 90.0% |
| | | 7 | 90.0% | 90.0% | 180.0% | 90.0% | 90.0% | 90.0% | 90.0% | 90.0% |
| | | 8 | 90.0% | 90.0% | 90.0% | 90.0% | 90.0% | 90.0% | 90.0% | 90.0% |
| | | 9 | 90.0% | 90.0% | 90.0% | 90.0% | 90.0% | 90.0% | 90.0% | 90.0% |

Table 7: Per-client Downstream Performance for 1041

| Partition | rank | client | FedCPQR | FedGCUR | K.-R. | K.-V. | L.-R. | L.-V. | R.-R. | R.-V. |
|---|---|---|---|---|---|---|---|---|---|---|
| IID | 10 | 0 | 15.6% | 6.6% | 10.5% | 10.5% | 10.5% | 10.5% | 10.2% | 11.1% |
| | | 1 | 15.6% | 6.5% | 10.5% | 10.5% | 10.5% | 10.7% | 10.2% | 11.1% |
| | | 2 | 15.6% | 6.2% | 10.5% | 10.5% | 10.5% | 10.5% | 10.2% | 11.1% |
| | | 3 | 15.6% | 6.8% | 10.5% | 10.5% | 10.5% | 10.5% | 10.2% | 11.1% |
| | | 4 | 15.6% | 6.6% | 10.5% | 10.5% | 10.5% | 10.5% | 10.2% | 11.1% |
| | | 5 | 15.6% | 7.5% | 10.5% | 10.5% | 10.5% | 10.5% | 10.2% | 11.1% |
| | | 6 | 15.6% | 6.5% | 10.5% | 10.5% | 10.5% | 10.7% | 10.2% | 11.1% |
| | | 7 | 15.6% | 6.3% | 10.5% | 10.5% | 10.5% | 10.5% | 10.2% | 11.1% |
| | | 8 | 15.6% | 6.3% | 10.5% | 10.5% | 10.5% | 10.7% | 10.2% | 11.1% |
| | | 9 | 15.4% | 6.2% | 10.5% | 10.5% | 10.5% | 10.5% | 10.2% | 11.1% |
| | 50 | 0 | 18.3% | 16.6% | 9.5% | 9.5% | 10.5% | 10.7% | 10.1% | 9.8% |
| | | 1 | 18.3% | 17.6% | 9.5% | 9.5% | 10.5% | 10.7% | 10.1% | 9.8% |
| | | 2 | 18.2% | 17.9% | 9.5% | 9.5% | 10.5% | 10.7% | 10.1% | 9.8% |
| | | 3 | 18.2% | 17.4% | 10.5% | 10.5% | 10.5% | 10.7% | 10.1% | 9.8% |
| | | 4 | 18.3% | 16.9% | 9.5% | 9.5% | 10.5% | 10.7% | 10.1% | 9.8% |
| | | 5 | 18.0% | 17.6% | 9.5% | 9.5% | 10.5% | 10.7% | 10.1% | 9.8% |
| | | 6 | 18.4% | 17.3% | 9.5% | 9.5% | 10.5% | 10.7% | 10.1% | 9.8% |
| | | 7 | 18.2% | 17.9% | 10.5% | 10.5% | 10.5% | 10.7% | 10.1% | 9.8% |
| | | 8 | 18.4% | 17.3% | 9.5% | 9.5% | 10.5% | 10.7% | 10.1% | 9.8% |
| | | 9 | 18.3% | 17.0% | 10.5% | 10.5% | 10.5% | 10.7% | 10.1% | 9.8% |
| | 100 | 0 | 15.7% | 21.0% | 10.8% | 10.8% | 13.7% | 10.1% | 14.6% | 11.2% |
| | | 1 | 15.0% | 20.3% | 10.8% | 10.8% | 13.8% | 10.1% | 14.4% | 11.2% |
| | | 2 | 16.1% | 20.7% | 10.5% | 10.5% | 13.5% | 10.1% | 14.4% | 11.1% |
| | | 3 | 15.6% | 20.6% | 10.8% | 10.8% | 14.0% | 10.1% | 14.4% | 11.2% |
| | | 4 | 15.7% | 20.7% | 10.8% | 10.8% | 13.7% | 10.1% | 14.6% | 11.1% |
| | | 5 | 15.3% | 20.6% | 10.8% | 10.8% | 13.8% | 10.1% | 14.6% | 11.2% |
| | | 6 | 15.6% | 21.0% | 10.8% | 10.8% | 14.0% | 10.1% | 14.6% | 11.1% |
| | | 7 | 16.1% | 20.7% | 10.8% | 10.8% | 13.5% | 10.1% | 14.4% | 11.0% |
| | | 8 | 16.0% | 21.2% | 10.7% | 10.7% | 14.0% | 10.1% | 14.3% | 11.1% |
| | | 9 | 16.0% | 21.2% | 10.5% | 10.5% | 13.8% | 10.1% | 14.4% | 11.1% |
| Non-IID | 10 | 0 | 9.5% | 9.9% | 10.5% | 10.5% | 10.5% | 10.7% | 10.2% | 11.1% |
| | | 1 | 9.4% | 11.5% | 10.5% | 10.5% | 10.5% | 10.7% | 10.2% | 10.5% |
| | | 2 | 9.4% | 8.8% | 10.5% | 10.5% | 10.5% | 10.7% | 10.2% | 10.5% |
| | | 3 | 12.1% | 8.5% | 10.5% | 10.5% | 10.5% | 10.5% | 10.2% | 10.5% |
| | | 4 | 9.5% | 9.8% | 10.5% | 10.5% | 10.5% | 10.7% | 10.2% | 11.1% |
| | | 5 | 9.2% | 9.4% | 10.5% | 10.5% | 10.5% | 10.5% | 10.2% | 10.5% |
| | | 6 | 9.2% | 9.8% | 10.5% | 10.5% | 10.5% | 10.5% | 10.2% | 11.1% |
| | | 7 | 8.9% | 9.5% | 10.5% | 10.5% | 10.5% | 10.5% | 10.2% | 11.1% |
| | | 8 | 9.4% | 9.7% | 10.5% | 10.5% | 10.5% | 10.5% | 10.2% | 11.1% |
| | | 9 | 9.2% | 9.9% | 10.5% | 10.5% | 10.5% | 10.7% | 10.2% | 11.1% |
| | 50 | 0 | 16.9% | 16.9% | 10.5% | 10.5% | 10.5% | 10.7% | 10.1% | 9.8% |
| | | 1 | 16.0% | 15.0% | 10.5% | 10.5% | 9.1% | 10.7% | 10.1% | 9.8% |
| | | 2 | 18.9% | 15.4% | 9.5% | 9.5% | 10.7% | 10.8% | 10.1% | 9.8% |
| | | 3 | 18.6% | 16.3% | 10.5% | 10.5% | 10.5% | 10.7% | 10.1% | 9.8% |
| | | 4 | 19.6% | 16.1% | 10.5% | 10.5% | 10.5% | 10.7% | 10.1% | 9.8% |
| | | 5 | 17.7% | 15.3% | 10.5% | 10.5% | 10.5% | 10.7% | 10.1% | 9.8% |
| | | 6 | 17.3% | 16.0% | 10.5% | 10.5% | 10.5% | 10.7% | 10.1% | 9.8% |
| | | 7 | 20.2% | 15.7% | 10.5% | 10.5% | 10.5% | 10.7% | 10.1% | 9.8% |
| | | 8 | 18.9% | 15.0% | 10.5% | 10.5% | 10.5% | 10.7% | 10.1% | 9.8% |
| | | 9 | 18.2% | 15.4% | 9.5% | 9.5% | 10.5% | 10.7% | 10.1% | 9.8% |
| | 100 | 0 | 16.6% | 19.0% | 10.5% | 10.5% | 13.5% | 10.1% | 14.7% | 11.2% |
| | | 1 | 14.8% | 19.0% | 10.8% | 10.8% | 14.1% | 9.9% | 14.7% | 11.2% |
| | | 2 | 14.7% | 18.0% | 11.7% | 11.7% | 15.1% | 10.1% | 14.7% | 12.4% |
| | | 3 | 16.0% | 18.6% | 12.1% | 12.1% | 15.1% | 10.1% | 15.0% | 12.4% |
| | | 4 | 14.3% | 20.3% | 10.7% | 10.7% | 13.5% | 10.1% | 14.7% | 11.2% |
| | | 5 | 16.0% | 19.6% | 11.2% | 11.2% | 14.4% | 10.1% | 14.7% | 12.0% |
| | | 6 | 17.1% | 20.3% | 10.5% | 10.5% | 13.5% | 10.1% | 14.7% | 11.2% |
| | | 7 | 15.9% | 20.6% | 10.5% | 10.5% | 14.1% | 10.1% | 14.7% | 11.4% |
| | | 8 | 14.3% | 19.0% | 10.5% | 10.5% | 13.7% | 10.1% | 14.7% | 11.2% |
| | | 9 | 15.4% | 19.7% | 10.7% | 10.7% | 14.1% | 10.1% | 14.7% | 11.2% |

Table 8: Per-client Downstream Performance for 40923

| Partition | rank | client | FedCPQR | FedGCUR | K.-R. | K.-V. | L.-R. | L.-V. | R.-R. | R.-V. |
|---|---|---|---|---|---|---|---|---|---|---|
| IID | 10 | 0 | 2.2% | 2.2% | 2.2% | 2.2% | 2.2% | 2.2% | 2.2% | 2.2% |
| | | 1 | 2.2% | 2.2% | 2.2% | 2.2% | 2.2% | 2.2% | 2.2% | 2.2% |
| | | 2 | 2.2% | 2.2% | 2.2% | 2.2% | 2.2% | 2.2% | 2.2% | 2.2% |
| | | 3 | 2.2% | 2.2% | 2.2% | 2.2% | 2.2% | 2.2% | 2.2% | 2.2% |
| | | 4 | 2.2% | 2.2% | 2.2% | 2.2% | 2.2% | 2.2% | 2.2% | 2.2% |
| | | 5 | 2.2% | 2.1% | 2.2% | 2.2% | 2.2% | 2.2% | 2.2% | 2.2% |
| | | 6 | 2.2% | 2.2% | 2.2% | 2.2% | 2.2% | 2.2% | 2.2% | 2.2% |
| | | 7 | 2.2% | 2.2% | 2.2% | 2.2% | 2.2% | 2.2% | 2.2% | 2.2% |
| | | 8 | 2.2% | 2.2% | 2.2% | 2.2% | 2.2% | 2.2% | 2.2% | 2.2% |
| | | 9 | 2.2% | 2.2% | 2.2% | 2.2% | 2.2% | 2.2% | 2.2% | 2.2% |
| | 50 | 0 | 2.5% | 3.0% | 2.2% | 2.2% | 2.2% | 2.2% | 2.2% | 2.2% |
| | | 1 | 2.5% | 3.0% | 2.2% | 2.2% | 2.2% | 2.2% | 2.2% | 2.2% |
| | | 2 | 2.5% | 3.0% | 2.2% | 2.2% | 2.2% | 2.2% | 2.2% | 2.2% |
| | | 3 | 2.5% | 3.0% | 2.2% | 2.2% | 2.2% | 2.2% | 2.2% | 2.2% |
| | | 4 | 2.5% | 3.0% | 2.2% | 2.2% | 2.2% | 2.2% | 2.2% | 2.2% |
| | | 5 | 2.6% | 3.0% | 2.2% | 2.2% | 2.2% | 2.2% | 2.2% | 2.2% |
| | | 6 | 2.5% | 3.0% | 2.2% | 2.2% | 2.2% | 2.2% | 2.2% | 2.2% |
| | | 7 | 2.5% | 3.0% | 2.2% | 2.2% | 2.2% | 2.2% | 2.2% | 2.2% |
| | | 8 | 2.5% | 3.0% | 2.2% | 2.2% | 2.2% | 2.2% | 2.2% | 2.2% |
| | | 9 | 2.5% | 3.0% | 2.2% | 2.2% | 2.2% | 2.2% | 2.2% | 2.2% |
| | 100 | 0 | 2.8% | 2.9% | 2.0% | 2.0% | 2.7% | 2.2% | 2.4% | 2.2% |
| | | 1 | 2.8% | 2.9% | 2.1% | 2.1% | 2.7% | 2.2% | 2.3% | 2.2% |
| | | 2 | 2.8% | 2.9% | 2.1% | 2.1% | 2.7% | 2.2% | 2.3% | 2.2% |
| | | 3 | 2.8% | 2.9% | 2.0% | 2.0% | 2.7% | 2.2% | 2.3% | 2.2% |
| | | 4 | 2.8% | 2.9% | 2.1% | 2.1% | 2.7% | 2.2% | 2.3% | 2.2% |
| | | 5 | 2.8% | 2.9% | 2.1% | 2.1% | 2.7% | 2.2% | 2.3% | 2.2% |
| | | 6 | 2.8% | 2.9% | 2.0% | 2.0% | 2.7% | 2.2% | 2.3% | 2.2% |
| | | 7 | 2.8% | 2.9% | 2.0% | 2.0% | 2.7% | 2.2% | 2.3% | 2.2% |
| | | 8 | 2.8% | 2.9% | 2.0% | 2.0% | 2.7% | 2.2% | 2.3% | 2.2% |
| | | 9 | 2.8% | 2.9% | 2.0% | 2.0% | 2.7% | 2.2% | 2.3% | 2.2% |
| Non-IID | 10 | 0 | 1.9% | 2.4% | 2.2% | 2.2% | 2.2% | 2.2% | 2.2% | 2.2% |
| | | 1 | 1.8% | 2.4% | 2.2% | 2.2% | 2.2% | 2.2% | 2.2% | 2.2% |
| | | 2 | 1.8% | 2.4% | 2.2% | 2.2% | 2.2% | 2.2% | 2.2% | 2.2% |
| | | 3 | 1.8% | 2.4% | 2.2% | 2.2% | 2.2% | 2.2% | 2.2% | 2.2% |
| | | 4 | 1.8% | 2.5% | 2.2% | 2.2% | 2.2% | 2.2% | 2.2% | 2.2% |
| | | 5 | 1.9% | 2.5% | 2.2% | 2.2% | 2.2% | 2.2% | 2.2% | 2.2% |
| | | 6 | 1.9% | 2.4% | 2.2% | 2.2% | 2.2% | 2.2% | 2.2% | 2.2% |
| | | 7 | 1.8% | 2.4% | 2.2% | 2.2% | 2.2% | 2.2% | 2.2% | 2.2% |
| | | 8 | 1.8% | 2.4% | 2.2% | 2.2% | 2.2% | 2.2% | 2.2% | 2.2% |
| | | 9 | 1.9% | 2.4% | 2.2% | 2.2% | 2.2% | 2.2% | 2.2% | 2.2% |
| | 50 | 0 | 2.7% | 3.0% | 2.2% | 2.2% | 2.2% | 2.2% | 2.2% | 2.2% |
| | | 1 | 2.5% | 3.0% | 2.2% | 2.2% | 2.2% | 2.2% | 2.2% | 2.2% |
| | | 2 | 2.5% | 2.9% | 2.2% | 2.2% | 2.2% | 2.2% | 2.2% | 2.2% |
| | | 3 | 2.5% | 2.9% | 2.2% | 2.2% | 2.2% | 2.2% | 2.2% | 2.2% |
| | | 4 | 2.5% | 2.9% | 2.2% | 2.2% | 2.2% | 2.2% | 2.2% | 2.2% |
| | | 5 | 2.5% | 2.9% | 2.2% | 2.2% | 2.2% | 2.2% | 2.2% | 2.2% |
| | | 6 | 2.5% | 2.9% | 2.2% | 2.2% | 2.2% | 2.2% | 2.2% | 2.2% |
| | | 7 | 2.5% | 3.0% | 2.2% | 2.2% | 2.2% | 2.2% | 2.2% | 2.2% |
| | | 8 | 2.5% | 3.0% | 2.2% | 2.2% | 2.2% | 2.2% | 2.2% | 2.2% |
| | | 9 | 2.5% | 2.9% | 2.2% | 2.2% | 2.2% | 2.2% | 2.2% | 2.2% |
| | 100 | 0 | 2.8% | 3.4% | 2.5% | 2.5% | 2.7% | 2.1% | 2.3% | 2.2% |
| | | 1 | 2.8% | 3.4% | 2.1% | 2.1% | 2.6% | 2.2% | 2.3% | 2.2% |
| | | 2 | 2.7% | 3.2% | 2.1% | 2.1% | 2.7% | 2.2% | 2.3% | 2.2% |
| | | 3 | 2.7% | 3.2% | 2.6% | 2.6% | 2.7% | 2.2% | 2.3% | 2.2% |
| | | 4 | 2.8% | 3.5% | 2.4% | 2.4% | 2.8% | 2.2% | 2.4% | 2.2% |
| | | 5 | 2.8% | 3.3% | 2.5% | 2.5% | 2.8% | 2.2% | 2.4% | 2.2% |
| | | 6 | 2.8% | 3.2% | 2.5% | 2.5% | 2.6% | 2.2% | 2.4% | 2.2% |
| | | 7 | 2.8% | 3.4%20 | 2.1% | 2.1% | 2.7% | 2.2% | 2.3% | 2.2% |
| | | 8 | 2.9% | 3.3% | 2.5% | 2.5% | 2.6% | 2.2% | 2.4% | 2.2% |
| | | 9 | 2.8% | 3.3% | 2.5% | 2.5% | 2.8% | 2.2% | 2.3% | 2.1% |

Table 9: Per-client Downstream Performance for 41082

| Partition | rank | client | FedCPQR | FedGCUR | K.-R. | K.-V. | L.-R. | L.-V. | R.-R. | R.-V. |
|---|---|---|---|---|---|---|---|---|---|---|
| IID | 10 | 0 | 19.0% | 21.2% | 16.3% | 16.3% | 20.3% | 16.1% | 29.4% | 26.0% |
| | | 1 | 19.1% | 21.3% | 16.3% | 16.3% | 20.1% | 16.1% | 28.0% | 26.0% |
| | | 2 | 19.1% | 21.3% | 16.2% | 16.2% | 20.1% | 16.2% | 28.0% | 25.6% |
| | | 3 | 18.9% | 21.4% | 16.0% | 16.0% | 20.1% | 16.2% | 28.4% | 26.0% |
| | | 4 | 19.2% | 21.6% | 16.0% | 16.0% | 20.2% | 16.2% | 28.7% | 25.9% |
| | | 5 | 18.9% | 20.9% | 16.0% | 16.0% | 20.2% | 16.2% | 29.1% | 25.8% |
| | | 6 | 19.2% | 21.5% | 15.9% | 15.9% | 20.2% | 16.2% | 28.7% | 26.1% |
| | | 7 | 19.1% | 21.1% | 16.1% | 16.1% | 20.1% | 16.2% | 29.4% | 26.0% |
| | | 8 | 19.3% | 21.2% | 16.0% | 16.0% | 20.1% | 16.2% | 29.2% | 25.9% |
| | | 9 | 19.0% | 21.1% | 16.1% | 16.1% | 20.1% | 16.2% | 29.2% | 26.0% |
| | 50 | 0 | 32.5% | 39.5% | 20.9% | 20.9% | 19.5% | 18.1% | 34.4% | 19.1% |
| | | 1 | 33.0% | 38.2% | 20.5% | 20.5% | 19.6% | 18.5% | 34.4% | 19.2% |
| | | 2 | 32.8% | 38.5% | 20.9% | 20.9% | 19.6% | 18.3% | 34.4% | 18.6% |
| | | 3 | 32.6% | 39.8% | 21.0% | 21.0% | 19.7% | 18.5% | 34.4% | 18.4% |
| | | 4 | 32.7% | 39.5% | 20.7% | 20.7% | 19.3% | 18.4% | 34.1% | 19.1% |
| | | 5 | 32.6% | 40.2% | 20.6% | 20.6% | 19.3% | 18.3% | 34.3% | 19.5% |
| | | 6 | 32.8% | 38.4% | 20.8% | 20.8% | 19.4% | 18.6% | 34.4% | 18.9% |
| | | 7 | 32.8% | 39.1% | 20.5% | 20.5% | 19.4% | 18.4% | 34.4% | 19.0% |
| | | 8 | 32.7% | 38.1% | 20.9% | 20.9% | 19.4% | 18.5% | 34.0% | 19.3% |
| | | 9 | 32.8% | 40.2% | 20.7% | 20.7% | 19.7% | 18.3% | 34.1% | 19.7% |
| | 100 | 0 | 38.3% | 48.8% | 24.0% | 24.0% | 36.8% | 30.9% | 37.4% | 46.1% |
| | | 1 | 38.6% | 48.8% | 23.9% | 23.9% | 36.9% | 30.8% | 37.0% | 46.4% |
| | | 2 | 38.1% | 48.7% | 24.1% | 24.1% | 36.8% | 30.6% | 37.2% | 46.5% |
| | | 3 | 38.2% | 48.8% | 24.3% | 24.3% | 37.0% | 30.5% | 37.4% | 46.6% |
| | | 4 | 38.5% | 48.4% | 23.8% | 23.8% | 36.6% | 30.4% | 37.0% | 46.1% |
| | | 5 | 38.3% | 49.1% | 24.4% | 24.4% | 36.7% | 30.4% | 36.9% | 46.2% |
| | | 6 | 38.0% | 48.8% | 24.0% | 24.0% | 37.0% | 30.5% | 37.7% | 46.3% |
| | | 7 | 38.0% | 49.0% | 24.1% | 24.1% | 36.7% | 30.5% | 37.2% | 46.2% |
| | | 8 | 38.2% | 48.9% | 24.1% | 24.1% | 36.8% | 30.5% | 36.8% | 46.1% |
| | | 9 | 38.1% | 49.1% | 24.2% | 24.2% | 36.9% | 30.6% | 37.0% | 46.3% |
| Non-IID | 10 | 0 | 34.2% | 37.4% | 17.5% | 17.5% | 20.8% | 17.8% | 29.2% | 25.1% |
| | | 1 | 33.1% | 40.3% | 17.7% | 17.7% | 21.3% | 20.4% | 29.9% | 25.6% |
| | | 2 | 33.6% | 40.3% | 16.3% | 16.3% | 20.4% | 16.1% | 30.5% | 26.5% |
| | | 3 | 35.1% | 38.4% | 16.2% | 16.2% | 19.7% | 16.3% | 25.5% | 25.6% |
| | | 4 | 32.5% | 40.4% | 16.8% | 16.8% | 20.5% | 16.3% | 29.0% | 26.2% |
| | | 5 | 31.7% | 39.4% | 16.6% | 16.6% | 18.3% | 17.6% | 24.5% | 25.5% |
| | | 6 | 33.0% | 40.8% | 16.4% | 16.4% | 20.6% | 16.5% | 26.5% | 25.3% |
| | | 7 | 32.0% | 38.8% | 19.1% | 19.1% | 19.4% | 16.1% | 28.7% | 24.8% |
| | | 8 | 32.2% | 39.9% | 17.3% | 17.3% | 19.8% | 16.5% | 30.5% | 25.4% |
| | | 9 | 32.3% | 39.4% | 17.8% | 17.8% | 20.3% | 18.6% | 29.6% | 25.7% |
| | 50 | 0 | 32.2% | 41.1% | 23.9% | 23.9% | 17.7% | 16.3% | 34.5% | 19.5% |
| | | 1 | 31.1% | 37.4% | 25.5% | 25.5% | 19.2% | 19.1% | 36.1% | 21.0% |
| | | 2 | 33.0% | 39.6% | 20.3% | 20.3% | 18.7% | 18.8% | 34.5% | 19.6% |
| | | 3 | 32.4% | 41.3% | 19.1% | 19.1% | 16.8% | 18.2% | 34.3% | 15.8% |
| | | 4 | 33.9% | 41.6% | 22.9% | 22.9% | 20.4% | 18.5% | 34.8% | 18.2% |
| | | 5 | 33.0% | 38.9% | 23.4% | 23.4% | 19.6% | 19.1% | 32.6% | 24.6% |
| | | 6 | 34.2% | 40.9% | 21.2% | 21.2% | 19.0% | 18.9% | 33.4% | 17.9% |
| | | 7 | 34.4% | 39.1% | 23.5% | 23.5% | 21.3% | 16.0% | 33.1% | 26.9% |
| | | 8 | 32.9% | 39.4% | 23.4% | 23.4% | 18.0% | 15.3% | 31.7% | 21.1% |
| | | 9 | 31.7% | 36.8% | 25.5% | 25.5% | 18.1% | 16.8% | 33.3% | 25.5% |
| | 100 | 0 | 37.9% | 40.2% | 24.5% | 24.5% | 34.7% | 29.8% | 38.2% | 45.2% |
| | | 1 | 38.8% | 38.3% | 24.8% | 24.8% | 33.3% | 32.0% | 40.2% | 45.8% |
| | | 2 | 38.0% | 41.2% | 23.9% | 23.9% | 34.9% | 28.8% | 37.6% | 45.4% |
| | | 3 | 38.8% | 40.8% | 23.8% | 23.8% | 33.9% | 27.4% | 39.0% | 45.8% |
| | | 4 | 39.8% | 41.7% | 24.6% | 24.6% | 36.2% | 30.4% | 38.2% | 43.5% |
| | | 5 | 38.7% | 39.9% | 24.5% | 24.5% | 37.6% | 29.6% | 36.9% | 46.2% |
| | | 6 | 40.1% | 42.7% | 23.2% | 23.2% | 35.3% | 29.1% | 36.5% | 47.6% |
| | | 7 | 39.2% | 39.7% | 22.1% | 22.1% | 36.5% | 26.9% | 34.5% | 45.5% |
| | | 8 | 38.6% | 40.6% | 23.0% | 23.0% | 34.4% | 30.6% | 35.8% | 45.3% |
| | | 9 | 36.2% | 38.1% | 24.5% | 24.5% | 36.5% | 31.1% | 36.2% | 41.5% |

Table 10: Per-client Downstream Performance for 41159

| Partition | rank | client | FedCPQR | FedGCUR | K.-R. | K.-V. | L.-R. | L.-V. | R.-R. | R.-V. |
|---|---|---|---|---|---|---|---|---|---|---|
| IID | 10 | 0 | 59.6% | 59.5% | 59.0% | 59.0% | 58.8% | 59.4% | 57.5% | 59.2% |
| | | 1 | 59.7% | 59.5% | 59.2% | 59.2% | 58.8% | 59.4% | 57.4% | 59.2% |
| | | 2 | 59.6% | 59.5% | 59.2% | 59.2% | 58.8% | 59.4% | 57.4% | 59.3% |
| | | 3 | 59.4% | 59.5% | 59.1% | 59.1% | 58.7% | 59.4% | 57.5% | 59.3% |
| | | 4 | 59.6% | 59.5% | 59.1% | 59.1% | 58.8% | 59.4% | 57.3% | 59.2% |
| | | 5 | 59.7% | 59.5% | 59.2% | 59.2% | 58.8% | 59.3% | 57.5% | 59.2% |
| | | 6 | 59.7% | 59.6% | 59.1% | 59.1% | 58.8% | 59.4% | 57.5% | 59.2% |
| | | 7 | 59.5% | 59.5% | 59.0% | 59.0% | 58.8% | 59.3% | 57.4% | 59.2% |
| | | 8 | 59.6% | 59.4% | 59.2% | 59.2% | 58.7% | 59.4% | 57.3% | 59.3% |
| | | 9 | 59.7% | 59.6% | 59.2% | 59.2% | 58.8% | 59.4% | 57.5% | 59.2% |
| | 50 | 0 | 59.8% | 58.9% | 58.7% | 58.7% | 58.2% | 58.7% | 54.2% | 55.0% |
| | | 1 | 59.9% | 59.1% | 58.8% | 58.8% | 58.2% | 58.8% | 53.4% | 54.8% |
| | | 2 | 59.9% | 59.0% | 58.7% | 58.7% | 58.0% | 58.7% | 53.8% | 54.5% |
| | | 3 | 59.8% | 59.0% | 58.7% | 58.7% | 58.2% | 58.6% | 53.3% | 54.3% |
| | | 4 | 59.9% | 58.9% | 58.6% | 58.6% | 58.3% | 58.8% | 53.6% | 54.7% |
| | | 5 | 59.9% | 58.9% | 58.9% | 58.9% | 58.2% | 58.7% | 53.7% | 54.5% |
| | | 6 | 59.8% | 59.0% | 58.9% | 58.9% | 58.3% | 58.8% | 54.4% | 55.0% |
| | | 7 | 59.8% | 58.9% | 58.7% | 58.7% | 58.1% | 58.7% | 53.3% | 54.3% |
| | | 8 | 59.9% | 58.9% | 58.7% | 58.7% | 58.2% | 58.7% | 53.9% | 54.5% |
| | | 9 | 59.8% | 59.1% | 58.8% | 58.8% | 58.3% | 58.8% | 54.1% | 55.1% |
| | 100 | 0 | 56.8% | 58.8% | 56.9% | 56.9% | 59.1% | 57.0% | 58.4% | 57.5% |
| | | 1 | 56.7% | 58.7% | 57.0% | 57.0% | 59.0% | 57.1% | 58.0% | 57.5% |
| | | 2 | 56.8% | 58.8% | 56.8% | 56.8% | 59.1% | 56.9% | 57.6% | 57.3% |
| | | 3 | 56.8% | 58.8% | 56.6% | 56.6% | 59.0% | 56.9% | 57.4% | 56.9% |
| | | 4 | 56.7% | 58.7% | 56.8% | 56.8% | 59.0% | 57.0% | 58.0% | 57.3% |
| | | 5 | 56.8% | 58.8% | 57.2% | 57.2% | 58.9% | 57.1% | 57.9% | 57.4% |
| | | 6 | 56.8% | 58.8% | 57.3% | 57.3% | 59.2% | 57.1% | 58.4% | 57.5% |
| | | 7 | 56.8% | 58.8% | 56.6% | 56.6% | 58.9% | 56.9% | 57.6% | 57.0% |
| | | 8 | 56.7% | 58.8% | 57.0% | 57.0% | 59.0% | 56.9% | 57.8% | 57.3% |
| | | 9 | 56.9% | 58.9% | 57.0% | 57.0% | 59.1% | 57.1% | 58.4% | 57.5% |
| Non-IID | 10 | 0 | 55.1% | 59.4% | 55.6% | 55.6% | 52.5% | 53.5% | 48.6% | 54.3% |
| | | 1 | 59.3% | 60.0% | 59.3% | 59.3% | 59.0% | 59.3% | 57.5% | 59.5% |
| | | 2 | 58.9% | 60.0% | 59.3% | 59.3% | 58.1% | 58.8% | 56.9% | 58.5% |
| | | 3 | 53.6% | 58.6% | 53.3% | 53.3% | 51.1% | 50.7% | 47.5% | 53.6% |
| | | 4 | 59.4% | 60.0% | 59.6% | 59.6% | 59.1% | 59.4% | 58.2% | 59.6% |
| | | 5 | 59.8% | 60.0% | 60.0% | 60.0% | 59.8% | 59.7% | 58.7% | 60.0% |
| | | 6 | 59.8% | 60.0% | 60.0% | 60.0% | 59.8% | 59.8% | 58.7% | 60.2% |
| | | 7 | 59.2% | 60.0% | 59.2% | 59.2% | 58.3% | 58.8% | 57.3% | 59.1% |
| | | 8 | 59.8% | 60.0% | 60.0% | 60.0% | 59.7% | 59.7% | 58.7% | 60.2% |
| | | 9 | 57.5% | 59.9% | 58.4% | 58.4% | 56.4% | 57.6% | 55.4% | 57.6% |
| | 50 | 0 | 55.6% | 60.0% | 53.1% | 53.1% | 52.1% | 54.7% | 45.9% | 49.4% |
| | | 1 | 59.8% | 60.0% | 59.4% | 59.4% | 58.4% | 58.9% | 54.4% | 55.8% |
| | | 2 | 59.4% | 60.0% | 57.9% | 57.9% | 56.8% | 58.1% | 52.1% | 53.7% |
| | | 3 | 54.2% | 59.9% | 50.7% | 50.7% | 51.0% | 53.9% | 45.3% | 48.7% |
| | | 4 | 59.6% | 60.0% | 59.6% | 59.6% | 58.8% | 59.1% | 55.3% | 56.4% |
| | | 5 | 60.0% | 60.0% | 59.8% | 59.8% | 59.2% | 59.7% | 56.9% | 57.4% |
| | | 6 | 60.0% | 60.0% | 59.9% | 59.9% | 59.5% | 59.7% | 57.9% | 57.7% |
| | | 7 | 59.8% | 60.0% | 58.7% | 58.7% | 57.1% | 58.4% | 52.4% | 54.2% |
| | | 8 | 60.0% | 60.0% | 59.9% | 59.9% | 59.4% | 59.8% | 57.8% | 57.7% |
| | | 9 | 58.0% | 60.0% | 57.2% | 57.2% | 55.5% | 57.5% | 49.6% | 52.5% |
| | 100 | 0 | 52.6% | 60.3% | 52.2% | 52.2% | 54.9% | 49.1% | 51.6% | 51.3% |
| | | 1 | 57.3% | 60.0% | 57.3% | 57.3% | 59.2% | 57.1% | 58.6% | 57.6% |
| | | 2 | 56.2% | 60.0% | 56.1% | 56.1% | 58.7% | 55.7% | 55.9% | 56.1% |
| | | 3 | 51.7% | 59.9% | 50.4% | 50.4% | 53.9% | 48.8% | 50.8% | 50.5% |
| | | 4 | 57.8% | 60.0% | 57.6% | 57.6% | 59.5% | 57.4% | 59.0% | 58.2% |
| | | 5 | 58.8% | 60.0% | 58.8% | 58.8% | 59.8% | 58.2% | 59.5% | 59.1% |
| | | 6 | 58.9% | 60.0% | 59.1% | 59.1% | 59.8% | 58.4% | 59.6% | 59.3% |
| | | 7 | 56.5% | 60.0% | 56.3% | 56.3% | 58.9% | 55.9% | 56.5% | 56.6% |
| | | 8 | 58.9% | 60.0% | 58.9% | 58.9% | 59.8% | 58.3% | 59.6% | 59.2% |
| | | 9 | 55.1% | 60.0% | 54.6% | 54.6% | 57.4% | 53.5% | 54.3% | 54.7% |

Table 11: Per-client Downstream Performance for 43985

| Partition | rank | client | FedCPQR | FedGCUR | K.-R. | K.-V. | L.-R. | L.-V. | R.-R. | R.-V. |
|---|---|---|---|---|---|---|---|---|---|---|
| IID | 10 | 0 | 2.9% | 8.7% | 6.8% | 6.8% | 6.1% | 11.3% | 6.3% | 9.6% |
| | | 1 | 2.9% | 8.7% | 6.7% | 6.7% | 6.1% | 11.3% | 6.2% | 9.4% |
| | | 2 | 2.9% | 8.8% | 6.6% | 6.6% | 6.0% | 11.3% | 6.4% | 9.5% |
| | | 3 | 2.9% | 8.7% | 6.5% | 6.5% | 6.2% | 11.7% | 6.0% | 9.3% |
| | | 4 | 2.9% | 8.7% | 6.7% | 6.7% | 6.1% | 11.3% | 6.1% | 9.5% |
| | | 5 | 3.1% | 8.7% | 6.6% | 6.6% | 6.0% | 11.5% | 6.1% | 9.6% |
| | | 6 | 2.9% | 8.5% | 6.6% | 6.6% | 6.2% | 11.4% | 6.0% | 9.4% |
| | | 7 | 2.9% | 8.7% | 6.7% | 6.7% | 5.9% | 11.2% | 6.4% | 9.4% |
| | | 8 | 3.0% | 8.7% | 6.5% | 6.5% | 6.2% | 11.3% | 6.2% | 9.6% |
| | | 9 | 3.0% | 8.7% | 6.7% | 6.7% | 6.0% | 11.3% | 6.3% | 9.2% |
| | 50 | 0 | 5.0% | 5.6% | 7.1% | 7.1% | 7.1% | 8.1% | 9.5% | 8.0% |
| | | 1 | 5.0% | 5.3% | 7.4% | 7.4% | 7.3% | 8.0% | 9.6% | 7.9% |
| | | 2 | 5.1% | 5.5% | 7.1% | 7.1% | 6.9% | 8.1% | 9.7% | 8.3% |
| | | 3 | 5.2% | 5.6% | 7.2% | 7.2% | 7.3% | 7.8% | 9.6% | 8.2% |
| | | 4 | 5.1% | 5.1% | 7.1% | 7.1% | 6.7% | 7.8% | 9.5% | 7.9% |
| | | 5 | 5.2% | 5.6% | 7.1% | 7.1% | 6.6% | 7.9% | 9.8% | 8.5% |
| | | 6 | 5.1% | 5.4% | 7.1% | 7.1% | 6.7% | 8.0% | 9.7% | 8.3% |
| | | 7 | 5.1% | 5.4% | 7.4% | 7.4% | 7.2% | 7.8% | 9.7% | 8.1% |
| | | 8 | 5.1% | 5.3% | 7.3% | 7.3% | 6.9% | 7.9% | 9.7% | 8.3% |
| | | 9 | 4.9% | 5.5% | 7.6% | 7.6% | 6.8% | 8.1% | 9.6% | 8.1% |
| | 100 | 0 | 10.9% | 5.8% | 9.0% | 9.0% | 10.3% | 9.1% | 9.3% | 9.3% |
| | | 1 | 11.0% | 6.1% | 8.7% | 8.7% | 10.1% | 9.2% | 9.4% | 9.5% |
| | | 2 | 11.0% | 6.0% | 9.2% | 9.2% | 10.3% | 9.0% | 9.4% | 9.4% |
| | | 3 | 11.1% | 6.0% | 9.2% | 9.2% | 9.9% | 9.0% | 9.4% | 9.3% |
| | | 4 | 11.0% | 6.0% | 8.9% | 8.9% | 10.1% | 8.9% | 9.6% | 9.4% |
| | | 5 | 11.0% | 5.9% | 9.3% | 9.3% | 10.0% | 9.1% | 9.3% | 9.2% |
| | | 6 | 11.0% | 6.0% | 9.2% | 9.2% | 10.3% | 9.1% | 9.4% | 9.4% |
| | | 7 | 11.0% | 5.9% | 8.8% | 8.8% | 9.7% | 9.1% | 9.4% | 9.5% |
| | | 8 | 11.0% | 5.7% | 9.2% | 9.2% | 10.1% | 9.2% | 9.3% | 9.2% |
| | | 9 | 11.0% | 5.9% | 9.2% | 9.2% | 10.1% | 9.4% | 9.4% | 9.6% |
| Non-IID | 10 | 0 | 5.1% | 6.9% | 6.9% | 6.9% | 6.2% | 12.9% | 6.2% | 8.9% |
| | | 1 | 5.9% | 7.8% | 6.5% | 6.5% | 7.4% | 10.1% | 6.8% | 8.4% |
| | | 2 | 5.9% | 8.5% | 6.6% | 6.6% | 5.8% | 10.9% | 6.3% | 8.3% |
| | | 3 | 5.4% | 7.4% | 7.0% | 7.0% | 6.1% | 11.1% | 6.2% | 9.2% |
| | | 4 | 5.6% | 6.7% | 6.1% | 6.1% | 6.2% | 11.3% | 6.6% | 10.4% |
| | | 5 | 5.4% | 7.9% | 6.3% | 6.3% | 6.2% | 10.4% | 6.7% | 9.4% |
| | | 6 | 5.1% | 7.4% | 8.3% | 8.3% | 6.2% | 10.6% | 6.0% | 8.8% |
| | | 7 | 5.2% | 7.6% | 6.7% | 6.7% | 5.9% | 11.7% | 6.2% | 10.3% |
| | | 8 | 5.2% | 7.6% | 6.2% | 6.2% | 6.0% | 11.2% | 6.3% | 9.9% |
| | | 9 | 4.9% | 7.0% | 7.1% | 7.1% | 6.0% | 12.1% | 6.5% | 9.2% |
| | 50 | 0 | 5.3% | 6.5% | 6.7% | 6.7% | 7.5% | 7.8% | 9.7% | 7.4% |
| | | 1 | 5.1% | 6.5% | 6.0% | 6.0% | 6.3% | 8.7% | 9.5% | 7.4% |
| | | 2 | 5.1% | 6.5% | 6.6% | 6.6% | 7.1% | 7.9% | 9.4% | 7.6% |
| | | 3 | 5.3% | 6.7% | 6.7% | 6.7% | 7.0% | 8.1% | 9.9% | 8.1% |
| | | 4 | 5.3% | 7.8% | 7.2% | 7.2% | 6.8% | 8.1% | 9.8% | 9.2% |
| | | 5 | 5.3% | 6.7% | 6.3% | 6.3% | 6.8% | 7.9% | 9.4% | 7.9% |
| | | 6 | 5.1% | 6.5% | 6.4% | 6.4% | 6.7% | 9.1% | 9.5% | 8.0% |
| | | 7 | 5.2% | 6.3% | 7.1% | 7.1% | 8.0% | 7.4% | 9.9% | 8.8% |
| | | 8 | 5.1% | 6.3% | 7.0% | 7.0% | 8.1% | 7.6% | 9.4% | 9.5% |
| | | 9 | 5.3% | 6.9% | 6.8% | 6.8% | 6.8% | 8.6% | 9.6% | 9.6% |
| | 100 | 0 | 11.0% | 5.6% | 8.8% | 8.8% | 9.3% | 9.9% | 9.5% | 9.0% |
| | | 1 | 11.0% | 6.0% | 7.6% | 7.6% | 8.9% | 9.5% | 10.3% | 9.0% |
| | | 2 | 11.7% | 6.0% | 8.7% | 8.7% | 9.1% | 9.2% | 9.6% | 9.2% |
| | | 3 | 10.9% | 5.8% | 8.6% | 8.6% | 10.3% | 9.6% | 9.2% | 9.0% |
| | | 4 | 10.6% | 7.0% | 8.9% | 8.9% | 10.3% | 9.1% | 9.2% | 9.6% |
| | | 5 | 10.8% | 6.5% | 7.9% | 7.9% | 8.9% | 8.7% | 9.6% | 10.1% |
| | | 6 | 10.3% | 4.9% | 7.4% | 7.4% | 10.3% | 8.7% | 9.6% | 9.0% |
| | | 7 | 10.4% | 5.6% | 238.9% | 8.9% | 10.8% | 8.9% | 8.5% | 9.2% |
| | | 8 | 10.9% | 5.4% | 9.3% | 9.3% | 10.5% | 9.1% | 8.3% | 10.2% |
| | | 9 | 11.0% | 5.6% | 9.1% | 9.1% | 9.7% | 9.2% | 9.1% | 9.6% |