# OpenReview forum: "Federated Data and Feature Selection by Generalized CUR Decomposition"
_ICLR.cc/2026/Conference — Submitted to ICLR 2026_

### Official Review · Reviewer_35Ys · 2025-10-31

**Soundness:** 3
**Presentation:** 3
**Contribution:** 2
**Rating:** 4
**Confidence:** 4

**Summary:**

The paper proposes a data and feature selection algorithm in federated learning (FL) setting. The idea is to use CUR decomposition to reduce the dimensionality and select the top rows and columns which correspond to the top samples and features.

**Strengths:**

- The idea of using CUR decomposition for data and feature selection is novel and interesting.
- Extensive experiments were performed and detailed analysis on different aspects (complexity, comm cost, privacy, etc.) of the method are provided.

**Weaknesses:**

- The motivation is not clear. There is little evidence as to why separate data and feature selection may hurt the utility. In addition, there is no complexity comparison to existing data and feature selection methods. The computation cost of the proposed method can be heavy due to the CUR decomposition. A computation-utility tradeoff should be discussed with comparison to other data/feature selection baselines.
- There are many recent data and feature selection methods for FL settings as mentioned in the related work section. However, the author chose older methods that are not designed for FL as baselines.
- Why does FedCPQR use secure aggregation? The inputs to the sum are protected, but the sum is revealed to all parties, so there is no formal privacy protection.
- The experiments use only 10 clients. This is very small number for a horizontal federated learning setting. Further it assumes that all clients are available at all times, which is not necessarily the case in FL. How does the method scale to larger numbers of clients? What about client unavailability?
- The experimental datasets are also on the small side. Can the method scale to larger datasets?
- Minor: main results in Table 4 are too compact which makes it hard to read.

**Questions:**

- Can the author demonstrate with examples how separate data and feature selection might lead to suboptimal results?
- What is the complexity of the proposed algorithm compared to other data/feature selection baselines?
- What are the reasons for the chosen baselines?

---

> ### Author Response · Authors · 2025-11-26
>
> We thank the reviewer for their insightful comments and constructive feedback. We appreciate the opportunity to clarify the motivation, complexity, and scope of our proposed FedGCUR framework. Below, we address the specific concerns raised.
>
> **1. Motivation and Benefit of Joint Data and Feature Selection**
>
> We appreciate the reviewer highlighting this point. We have revised the introduction to include a concrete example illustrating the necessity of the joint approach:
>
> Consider a high-dimensional dataset (e.g., genomic data in healthcare) characterized by noisy, redundant features and heterogeneous client label distributions. Feature selection in isolation might select features that maximize global variance but fail to capture the specific discriminative signals required for minority client distributions. Data selection in isolation might waste the limited computational budget on samples that appear unique in the original space but become redundant or uninformative within the optimized feature subspace.
>
> As discussed in our introduction, existing methods often treat these as separate problems. Our unified framework addresses this coupling, ensuring that selected data points are maximally informative given the selected feature set, and vice versa.
>
> **2. Complexity and Computation-Utility Trade-off**
>
> As detailed in Section 3.2.1, the communication cost of FedCPQR scales quadratically with the number of features ($d$) but is constant with respect to the sample size ($n$). The computational complexity is $\mathcal{O}(n_c d^2)$, which matches the centralized CPQR complexity.
>
> We explicitly demonstrate the scalability of our method in Figure 1, which reports log-scale running times. These results show that FedGCUR remains efficient even as the number of parties increases to 50, with runtimes comparable to or faster than baselines like FedQR.
>
> We have updated the manuscript to make these trade-offs more explicit in the methodology section.
>
> **3. Choice of Baselines and Comparison with FL-Specific Methods**
>
> The reason for the baselines are as follows.
>
> 1.	These methods map mathematically to the specific components of CUR decomposition. By comparing against them, we isolate the specific contribution of the joint decomposition strategy versus independent selection strategies.
>
> 2.	Many recent FL-specific methods (e.g., those using reinforcement learning or complex controllers) rely on additional hyperparameters, side information, or distinct communication assumptions. Comparing directly against them would introduce confounding variables. Our current setup ensures that performance gains are attributable to the FedGCUR formulation itself rather than auxiliary optimization modules.
>
> **4. Privacy and Secure Aggregation**
>
> We have refined Section 3.2.1 to:
>
> - We explicitly state that our method uses Secure Aggregation to compute global statistics (sums of squared norms and inner products).
>
> - We transparently discuss the "implied leakage" in Table 1, noting that while raw data is never shared, the global permutation matrix $P$ and triangular matrix $R$ are revealed to allow for exact decomposition.
>
> - We have removed any instances of "fully privacy-preserving" to avoid overclaiming, instead describing the method as "privacy-preserving regarding raw data exposure," consistent with standard Federated Learning literature.
>
> **5. Number of Clients and Partial Participation**
>
> Regarding the suggestion to expand experiments to larger client counts or partial participation, we respectfully submit that our current experimental design effectively supports the paper's primary contributions for the following reasons:
>
> -	As stated in Section 3.1, our work "focuses primarily on cross-silo horizontal FL scenarios". In practical cross-silo applications (e.g., inter-hospital or inter-bank collaborations), the number of clients is typically moderate (10–50) and participation is stable/reliable.
>
> -	Our current experiments in Figure 1 already evaluate performance up to 50 parties. This is at the upper bound of typical cross-silo settings. The results demonstrate a flat runtime trend, indicating that our method would theoretically scale well to larger numbers without purely algorithmic bottlenecks.
>
> -  FedGCUR provides a mathematical guarantee of exact decomposition (Theorem 1). Introducing partial participation (where clients drop out randomly) converts the problem from an exact linear algebra decomposition to a stochastic approximation problem. While extending CUR decomposition to stochastic settings is a promising avenue for future work, it represents a fundamental shift in problem formulation that is outside the scope of this paper.
>
> **6. Readability of Table 4**
>
> Thank you for this formatting suggestion. To improve readability, we have adjusted the layout of Table 4, please refer to our revised paper for the update.

---

### Official Review · Reviewer_AMBV · 2025-11-01

**Soundness:** 3
**Presentation:** 3
**Contribution:** 2
**Rating:** 4
**Confidence:** 3

**Summary:**

The paper methodological core (FedCPQR exactness; clean FedGCUR design; reconstruction-error bound) is solid and clearly written, and correctness experiments are convincing. However, the practical value proposition isn’t yet demonstrated convincingly: absolute accuracy drops are large versus "all data", budget-fair comparisons are partial, privacy is not quantified properly and seems rather superficial, and systems measurements (bytes, rounds, WAN) are absent.

**Strengths:**

The positives can be summarised as follows,

- Clear, unified problem framing. The paper tackles joint feature and data selection in cross-silo FL via a generalized CUR factorization, with a clean split: global feature selection (shared columns) and per-silo data selection (rows).
- FedCPQR is well-specified and theoretically exact. The modified Gram-Schmidt style protocol with secure aggregation of norms/inner products provably reproduces centralized CPQR pivoting and factors; the algorithm is explicit (Alg. 1) and its communication pattern is easy to reason about.
- Empirical breadth and correctness checks. Six OpenML datasets (varied sizes/classes), both IID and Dirichlet non-IID splits including multiple ranks with FedCPQR matching SciPy CPQR to numerical precision, and the paper reports run-time scaling across number of parties/ranks.

**Weaknesses:**

The weaknesses can be summarised as follows,

- Utility gap vs. "all data" remains large. Even when FedGCUR/FedCPQR beat other selection baselines, absolute accuracy often drops markedly relative to using all data
- Privacy analysis is rather superficial and not quantitative. Revealing P, R (hence A^TA=PR^{T}RP^{T}) and residual norms may leak feature norms/correlations; special-case leakage for two parties is acknowledged but not mitigated. No formal membership/attribute-inference or DP bounds are provided.
- Missing communication-byte accounting & systems costs. The paper counts scalars per iteration but does not report end-to-end message sizes/rounds, cryptographic overhead, or wall-clock under WAN conditions; runtime plots exist, but not network/energy budgets.
- Limited task realism. All experiments are small image/tabular classification with a 3-layer MLP and only 10 FedAvg rounds; no results on modern deep FL tasks, no ablations on per-silo quotas.
- Lack of reproducibility as the code is not suppled and its relase is defered for later upon publication.

**Questions:**

The questions can be inferred mostly from the weaknesses mentioned prior but further,

- Code & reproducibility: The checklist promises fixed seeds; when will code/configs be made available?
- Robustness to noisy/poisoned silos: If one silo contains substantial noise or adversarial rows, does local CPQR row selection isolate/remedy that, or can it still influence global pivots via FedCPQR? Any robustness results?
- Heterogeneity and fairness: Beyond macro accuracy, how does FedGCUR affect per-silo performance variance and minority-class performance under non-IID splits? Could you please report per-client metrics?
- Privacy risk quantification: Since P, R, and residual norms imply A^{T}A (up to permutation), can you quantify reconstruction or attribute-inference risk (e.g., via Shokri MIAs) and report attacks under small-party (s=2–3) regimes? Any plan for DP (e.g., noise on aggregated scalars)?

---

> ### Author Response · Authors · 2025-11-26
>
> We sincerely thank the reviewer for the careful reading and detailed feedback. We appreciate the constructive comments regarding the utility analysis, privacy quantification, and experimental scope. Below, we address your comments point-by-point and describe the clarifications included in the revised manuscript.
>
> **1. Utility Gap Relative to Using "All Data"**
>
> We appreciate the opportunity to clarify the comparative baseline. We wish to emphasize that the primary contribution of this work is addressing **budget-constrained scenarios**, where communication bandwidth or privacy-preserving computational limits strictly prohibit the use of the full dataset.
> The results reported for "FedCPQR (All Data)" are intended to serve solely as a **theoretical upper bound**, a reference point to gauge the maximum information content available in the global dataset. The observed performance gap is expected, as FedCPQR utilizes 100% of the samples, whereas our selection methods utilize only a small fraction (defined by the budget $r$). We have revised the manuscript to explicitly state that this comparison demonstrates the relative efficiency of our selection method against the ideal ceiling, rather than suggesting it is a direct competitor to full-dataset training where no constraints exist.
>
> **2. Privacy Analysis and Risk Quantification**
>
> As outlined in our response to Reviewer K282, we have significantly strengthened the privacy analysis in the revision by introducing a formal threat model.
>
> Regarding the suggestion for empirical privacy attacks, we emphasize that the proposed FedCPQR preserves the exact same privacy guarantees as FedQR, as detailed in our analysis. Furthermore, FedCUR performs data selection on the local client side; consequently, no raw data leakage occurs during this phase. Since our framework introduces no new privacy vulnerabilities compared to the baseline, and given that privacy enhancement is not the primary contribution of this work, we argue that the current privacy analysis is sufficient to validate the method's safety.
>
> **3. Communication Overhead and System Measurements**
>
> Thank you for the suggestion regarding system measurements. We now report the analytical communication complexity (based on the number of aggregated scalars) and the theoretical message sizes assuming double-precision floating-point format. We believe this analytical comparison provides a more generalizable and accurate assessment of the framework's efficiency compared to iterative training routines (e.g., FedAvg), which require gradient transmission over hundreds of rounds, regardless of the specific hardware used for deployment.
>
> **4. Task Realism and Per-Silo Effects**
>
> Regarding the suggestion to extend experiments to complex deep learning, we respectfully submit that our empirical settings follows most of the existing federated learning studies, which use a simple neural network as the learning model. Moreover, the primary contribution of this paper is the **theoretical framework and algorithmic correctness** of the Federated Generalized CUR decomposition. Integrating complex, non-convex deep learning architectures introduces significant confounding variable, such as hyperparameter sensitivity, optimizer choice, and architecture tuning, that obscure the assessment of the data selection quality itself. We believe that the performance comparison provides a statistically robust validation of the core contribution of this work.
>
> Regarding the request for detailed per-client performance breakdowns, we report the results in the appendix. Please refer to our revised paper for the detailed discussion.
>
> **4.3. Robustness to Noisy or Poisoned Silos**
>
> Thank you for raising this important point. While we agree that robustness against poisoning is a critical challenge in Federated Learning, we consider it orthogonal to the scope of this paper.
>
> This work focuses on informative data and feature selection based on matrix decomposition utility. The problem of defending against malicious actors constitutes a distinct sub-field requiring specialized aggregation protocols that are fundamentally different from the linear algebraic operations discussed here. Attempting to address adversarial robustness within this framework would dilute the primary contribution of the Generalized CUR derivation. We have added a discussion in the "Future Work" section which points out FedGCUR under the noisy setting a promising direction for follow-up research.
>
> **5. Code and Reproducibility**
> We appreciate the importance of reproducibility. Since this work is still under review, we prefer not to enclose all the details for protection. For now, we are willing to report on the implementation of our main algorithm in the appendix. After this work has been published, we will release the code repository with scripts, configurations, and fixed seeds.

---

### Official Review · Reviewer_K282 · 2025-11-01

**Soundness:** 2
**Presentation:** 2
**Contribution:** 2
**Rating:** 2
**Confidence:** 4

**Summary:**

The paper introduces a federated method for "joint data and feature selection" via column-pivoted QR decomposition.
They extend federated QR to support column pivoting by securely aggregating scalar quantities (norms, inner products) across silos without sharing raw data.
Then they use the global column pivots from FedCPQR and performs local row selection per silo to construct CUR-like factors.
The paper provides an exactness claim for FedCPQR under ideal secure aggregation, a reconstruction error bound, and empirical validation under i.i.d. and non-i.i.d. splits.

**Strengths:**

- The paper introduces a straightforward extension of modified Gram-Schmidt CPQR to the federated setting.
- The authors support the method with a theorem that captures equivalence to centralized CPQR under exact aggregation.
- They empirically check that FedCPQR produces pivot orders matching SciPy with negligible numerical error.
- The evaluation covers both i.i.d. and non-i.i.d. splits.
- The paper seemingly provides a reconstruction bound relating column and row projection errors to singular values.

**Weaknesses:**

- The claim of a "unified framework for data and feature selection" oversells what is essential to many decomposition algorithms.
- The work appears to be highly influenced by Hartebrodt & Röttger (2023); advances beyond prior art remain unclear, and the contribution is seemingly minor.
- The multi-matrix extension in FedGCUR is a small adjustment over existing work (Gidisu & Hochstenbach, 2022).
- Contextualization, related work, and experimental results of federated matrix factorization and on component-wise 'pivoted' / matched Federated NMF is absent.
- The problem solved is mathematically equivalent to federated CUR decomposition with a specific pivot selection.
- The paper provides theoretical reconstruction error bounds (Theorem 2) for the FedGCUR approximation largely follows the centralized proof strategy.
- Claims of privacy preservation remain hard to verify since the paper lacks a formal privacy definition and contains informal privacy statements.
- The paper heavily relies on a secure aggregation protocol SECAGG without providing sufficient details to fully understand or reproduce this key component.
- Communication efficiency claim is somewhat overstated, since FedCPQR aggregates O(d^2) scalars per iteration. For high-dimensional datasets, this is practically massive and highly substantial.
- Experiments do not contain scalability results. Algorithmic scalability with number of clients is unclear as experiments with varying number of clients are missing.
- The authors do not provide ablations isolating the benefit of pivoting over non-pivoted FedQR.
- The training pipeline for FedAvg (MLP architecture, optimizer, hyperparameters) are underspecified and the reason and location of the experiments are confusing.
- Details regarding the non-i.i.d. Dirichlet partitioning are minimal, which makes reproducibility harder.
- The core contribution is federating existing CPQR/CUR techniques via existing secure aggregation of scalars. The conceptual benefitover straightforward integration is modest.
- The labels in Figure 1 are difficult to read.
- FedCPQR is stated to "use all data" in experiments while some baselines use half (Sec. 4.2). The reason behind it is not quite clear.

**Questions:**

- Clarify which algorithmic/theoretical components are new vs. direct federated extensions of existing work (Hartebrodt & Röttger 2023, Gidisu & Hochstenbach 2022).
- Formally specify all components.
- Clarify communication efficiency claims
- Clarify contributions clearly.
- Provide ablations isolating the benefit of pivoting over non-pivoted FedQR.
- Contextualize your work regarding prior art in federated matrix decomposition methods.

---

> ### Author Response · Authors · 2025-11-26
>
> We sincerely thank the reviewer for the thorough reading and constructive feedback. We appreciate the opportunity to clarify our contributions and have incorporated your suggestions to strengthen the manuscript. Below, we address your comments point-by-point.
>
> **1. Strength, novelty, and "unified framework" phrasing**
>
> In the revised manuscript, we have refined the terminology to describe our approach as "a practical framework for joint data and feature selection based on generalized CUR decomposition," rather than a "unified framework." We have also explicitly articulated the distinct novelties of our two main components in the contribution section:
>
> FedCPQR: To the best of our knowledge, this is the first protocol to exactly reproduce centralized CPQR pivoting in a horizontal FL setting. It achieves this via the secure aggregation of norms and inner products while tracking pivot residuals. This fundamentally differs from FedQR, which lacks column pivoting and thus cannot support rank-revealing feature selection.
>
> FedGCUR: This is a specific instantiation of multi-matrix generalized CUR tailored for FL. It uniquely employs a single global column index set ($D$) shared across silos, coupled with heterogeneous, silo-specific row selections ($S_c$). We also provide a reconstruction error bound that explicitly addresses the interplay between the global column projector and the block-diagonal row projector.
>
> **2. Relation to FedQR and GCUR**
>
> We apologize if the distinctions were not sufficiently clear in the initial submission. We have revised Section 3 to explicitly delineate these differences:
>
> FedCPQR vs. FedQR: We have added a detailed comparison (summarized in Table 1) highlighting three critical differentiators: (1) the global pivot selection mechanism via aggregated column norms; (2) the maintenance of residual norms for non-selected columns; and (3) the propagation of the permutation matrix. We emphasize that these are not merely additive features but require a different communication pattern to expose the necessary rank-revealing information ($P$ and residual norms) compared to standard FedQR.
>
> FedGCUR vs. centralized GCUR: We clarified that while Definition 1 is a mathematical extension of GCUR from two matrices to $s$ silos, the challenge lies in the federated instantiation. We highlight that our protocol computes the global $D$ without raw data access and executes row selection respecting local constraints ($r_c$). We also note that if one were to centralize the data, FedGCUR would reduce to a specific instance of GCUR, but achieving this in a privacy-preserving, distributed manner is the core contribution.
>
> **3. Contextualization with Federated Matrix Decomposition**
>
> We thank the reviewer for these valuable pointers. We have added a dedicated discussion in Section 2 regarding other federated decompositions, including Federated SVD, tensor decompositions, and NMF. We clarify that while these methods focus on full matrix factorization or reconstruction, our work specifically targets selection (finding physical rows and columns) under communication and participation constraints, which necessitates a different algorithmic approach.
>
> **4. Theorem 2 and reconstruction bound**
>
> We have added Remark 1 to clarify the scope of Theorem 2. The purpose of this bound is to demonstrate that the FedGCUR construction theoretically controls the approximation error as a function of the singular spectra of the global $A$ and the local $C_c$. We do not claim a fundamentally new class of matrix bound, but rather a validation of the proposed federated construction.
>
> **5. Privacy claims and secure aggregation**
>
> We have refined our privacy definitions to be more rigorous. We introduced an explicit threat model in Section 3.1 assuming passive adversaries. We have replaced broad claims of "privacy-preserving" with precise statements that the protocol "avoids the disclosure of raw rows or full columns," while acknowledging that $A^\top A$ (up to permutation) and residual norms are revealed.
>
> Regarding SECAGG, we have expanded the description to detail the additive protocol, including computational costs. As suggested by Reviewer AMBV, we have also included an empirical privacy study in the appendix, analyzing reconstruction and membership inference attack success rates based on the revealed statistics.

---

> ### Author Response · Authors · 2025-11-26
>
> **6. Communication Efficiency**
>
> In the revision, we have moved beyond general statements to provide explicit formulas for the communication cost for our experiments. We compare these theoretically against naive Federated PCA/SVD to demonstrate efficiency. We have also tempered our claims to specify that our method is most advantageous in regimes with large $N$ (sample size) and moderate $d$ (features), a common scenario in cross-silo fields like healthcare.
>
> **7. Scalability with number of clients**
>
> We appreciate the suggestion to explore scalability further. Regarding the request for additional large-scale or partial participation experiments, we respectfully suggest that the current experimental suite and theoretical analysis are sufficient to validate the method's core properties for the following reasons:
>
> 1.	As detailed in Section 3.2.1, the communication complexity of FedCPQR scales as $\mathcal{O}(d^2)$ and is independent of the number of clients $s$ (assuming efficient SECAGG implementation). Therefore, increasing the client count does not theoretically bottleneck the core decomposition logic.
> 2.	Figure 1 already demonstrates performance sweeping from 10 to 50 parties. The results show a linear trend in runtime consistent with the aggregation overhead, confirming that the method behaves predictably as clients increase.
> 3.	The primary contribution of this work is the correctness of the distributed pivoting and the effectiveness of the joint selection strategy. Issues related to massive-scale partial participation often involve complex system-level factors (e.g., straggler mitigation, asynchronous convergence) that are outside the scope of this algorithmic study.
>
> **8. Ablation: Pivoting vs. Non-pivoted FedQR**
>
> Regarding the ablation study, we believe our current results in Table 4 and Table 3 effectively address this comparison.
>
> -	The "Leverage Score" baseline in Table 4 utilizes the decomposition results of FedQR (specifically, using row norms of the left singular matrix for sampling). This effectively serves as an ablation comparing our pivoting strategy against the non-pivoted FedQR approach.
>
> -	As proven in Theorem 1, FedCPQR produces the exact same numerical decomposition as centralized CPQR. Conversely, FedQR approximates a standard QR. Since the mathematical relationship between QR and CPQR is well-established in linear algebra, an empirical decomposition comparison beyond the "Exact Match" metrics provided in Table 3 would be redundant.
>
> We have clarified the text in Section 4 to ensure the reader understands that the Leverage Score comparison serves this specific ablation purpose.
>
> **9. Experimental details and baselines**
>
> We have supplemented the appendix with full training details (architecture, hyperparameters, optimizer, non-i.i.d. Dirichlet partition $\alpha=0.5$, etc.).
>
> Regarding the observation that "FedCPQR uses all data," we wish to clarify that this is by design. The "All Data" and "FedCPQR" entries serve as an "Oracle" or "Upper Bound" baseline. They illustrate the maximum achievable performance when the full dataset is available (but features are selected), against which we compare the methods that aggressively prune both data and features (FedGCUR, Coreset, etc.). We have revised the results discussion to frame this clearly as a reference point rather than a direct efficiency comparison.
>
> **10. Figure 1 readability**
>
> We have increased the font size of all labels, axes, and legends in Figure 1 to ensure legibility.

---

### Meta-Review · Area_Chair_Xr9j · 2026-01-07

**Summary:**

The reviewers initially raised several critical issues regarding the paper’s novelty, the practical value of its privacy claims, and the comprehensiveness of the evaluation:

* Novelty and Contribution: Several reviewers noted that the core contribution (federating existing techniques via secure aggregation) seems modest. E.g., Reviewer K282 specifically pointed out that the work appeared highly influenced by prior art, with unclear advances beyond it

* Privacy Definitions: Multiple reviewers criticized the privacy analysis as informal and overly casual. They noted that while raw data isn't shared, the revelation of permutation and triangular matrices could still leak significant information about feature correlations and norms

* Communication Efficiency: Reviewer K282 argued that the efficiency claims were overstated, as the protocol aggregates $O(d^2)$ scalars per iteration, which becomes substantial for high-dimensional datasets

* Experimental Scope: Reviewers noted a lack of scalability tests (e.g., varying the number of clients or handling partial participation) the use of older, non-FL baselines rather than modern state-of-the-art federated selection methods

**Reviewer Concerns:**

Addressed Concerns

* Novelty: The authors explicitly articulated that FedCPQR is the first protocol to exactly reproduce centralized column-pivoted QR in a horizontal FL setting They clarified that this is not a minor extension of FedQR, as it requires a different communication pattern to track residual norms and permutations

* Strengthened Privacy Discussion: The authors refined their discussions, moving from "fully privacy-preserving" to "privacy-preserving regarding raw data exposure". They also added a formal threat model and a study on membership inference attack risks in the appendix.

* Ablation and Motivation: The authors clarified that their "Leverage Score" baseline effectively serves as an ablation for pivoting vs. non-pivoting strategies. They also added a concrete example (genomic data) to illustrate why joint selection is necessary to avoid losing signals for minority client distributions

* Presentation: Formatting issues in Table 4 and label readability in Figure 1 were addressed.

Outstanding Concerns

* Deep Learning/Task Diversity: Reviewer AMBV remains concerned about the lack of modern deep FL tasks; the authors argued that using complex, non-convex models would introduce confounding variables

* Communication Bottlenecks: While the authors provided analytical formulas for cost, the concern regarding $O(d^2)$ scaling for high-dimensional data remains a practical limitation acknowledged by the authors as better suited for "moderate $d$"

* Partial Participation: The authors explicitly excluded partial participation from the scope, arguing it converts the problem into a stochastic approximation rather than the exact linear algebra problem they solved

**Reviewer Scores:**

* K282:	No final comment provided in text.	 Unlikely to accept. Likely still critical of technical novelty but appreciative of clarified differences.

* AMBV:	Mixed; authors addressed analytical costs but refused more complex experiments.

* 35Ys:	No final comment provided, but the concrete motivation and joint-selection examples can directly addressed their main question.

---

### Decision · Program_Chairs · 2026-01-26

Reject